# Strain-restricted transfer of ferromagnetic electrodes for constructing reproducibly superior-quality spintronic devices

Lidan Guo [1,2,11], Xianrong Gu[1,11], Shunhua Hu[1,2,11], Wenchao Sun[3], Rui Zhang[1,4], Yang Qin[1], Ke Meng[1,2], Xiangqian Lu[5], Yayun Liu[1], Jiaxing Wang[4], Peijie Ma[4], Cheng Zhang [1], Ankang Guo[1,6], Tingting Yang[1,2], Xueli Yang[1,6], Guorui Wang[7], Yaling Liu[1], Kai Wang [8], Wenbo Mi [3], Chuang Zhang [6], Lang Jiang [6], Luqi Liu [1], Kun Zheng [4], Wei Qin[5] ✉, Wenjing Yan[9] & Xiangnan Sun [1,2,10] ✉

Spintronic device is the fundamental platform for spin-related academic and practical studies. However, conventional techniques with energetic deposition or boorish transfer of ferromagnetic metal inevitably introduce uncontrollable damage and undesired contamination in various spin-transport-channel materials, leading to partially attenuated and widely distributed spintronic device performances. These issues will eventually confuse the conclusions of academic studies and limit the practical applications of spintronics. Here we propose a polymer-assistant strain-restricted transfer technique that allows perfectly transferring the pre-patterned ferromagnetic electrodes onto channel materials without any damage and change on the properties of magnetism, interface, and channel. This technique is found productive for pursuing superior-quality spintronic devices with high controllability and reproducibility. It can also apply to various-kind (organic, inorganic, organic-inorganic hybrid, or carbon-based) and diverse-morphology (smooth, rough, even discontinuous) channel materials. This technique can be very useful for reliable device construction and will facilitate the technological transition of spintronic study.

Spintronics has attracted considerable attention in past decades because it can employ the electron spin to deal with information storage, transport, and processing in a high-efficiency way[1–3]. In this emerging field, information propagation normally relies on the spin-polarized current injection from ferromagnetic metals (FMs) and then transport in non-magnetic material (NMM) channels in spintronic devices[4–6]. The stable and reproducible spintronic device composed of FMs and NMMs is considered as the fundamental platform for realizing

[1]Key Laboratory of Nanosystem and Hierarchical Fabrication, CAS Center for Excellence in Nanoscience, National Center for Nanoscience and Technology, 100190 Beijing, People's Republic of China. [2]Center of Materials Science and Optoelectronics Engineering, University of Chinese Academy of Sciences, 100049 Beijing, People's Republic of China. [3]School of Science, Tianjin University, 300072 Tianjin, People's Republic of China. [4]Beijing Key Laboratory of Microstructure and Property of Solids, Faculty of Materials and Manufacturing, Beijing University of Technology, 100124 Beijing, People's Republic of China. [5]School of Physics, State Key Laboratory of Crystal Materials, Shandong University, 250100 Jinan, People's Republic of China. [6]Institute of Chemistry, Chinese Academy of Sciences, 100190 Beijing, People's Republic of China. [7]CAS Key Laboratory of Mechanical Behavior and Design of Materials, Department of Modern Mechanics, University of Science and Technology of China, 230027 Hefei, People's Republic of China. [8]Key Laboratory of Luminescence and Optical Information, Ministry of Education, Beijing Jiaotong University, 100044 Beijing, People's Republic of China. [9]School of Physics & Astronomy, University of Nottingham, Nottingham NG7 2RD, UK. [10]School of Material Science and Engineering, Zhengzhou University, 450001 Zhengzhou, People's Republic of China. [11]These authors contributed equally: Lidan Guo, Xianrong Gu, Shunhua Hu. ✉e-mail: wqin@sdu.edu.cn; sunxn@nanoctr.cn

the spin functionalities and studying the deep spin-related physical mechanisms in this field, which requires both damage-free FMs magnetism[7,8] and controllable FM/NMM interface[9] simultaneously. However, recent studies have demonstrated that metal deposition processes using energetic ways frequently introduce defects, stress, disorder, and metal diffusion into NMMs-based electronic devices[5,10,11], partially degrading the device performances and leading to poor reproducibility. Worse, this problem will deteriorate sharply in spintronic devices since FMs with very high melting points should be normally deposited in a much higher energetic way. This can surely inhomogeneously change the interfacial profile and interior magnetic-domain properties of FM electrodes simultaneously. More seriously, these problems are far beyond precise control and will lead to randomly varying spin injection-transport processes and wide distribution in device performances, which will finally confuse the consequences of academic studies and also limit the practical applications in the spintronics.

Electrode-transfer technique[12–16], relying on pre-deposition and physical transfer of metallic patterns onto channel materials, has been proposed as one best methods to completely avoid the above-described damage from metal deposition during electronic device preparation. However, such an approach can hardly be realized for fabricating reproducible spintronic devices because of the serious challenges as follows. First of all, the magnetic property of the FMs that is closely related to its microstructures can hardly stay constant after the electrode-transfer process, since it is quite sensitive to even a very tiny strain history in this process[17–20]. This requires that the strain on any points of FM electrodes and at any time during the transfer process should be less than its ultimate strain of remaining their magnetic properties (normally a few percent)[21,22]. Furthermore, during the transfer process, any introduced impurities at interfaces (e.g., oxygen[16,23,24], $H_2O$[16], and so on, maybe harmless to carrier transport) can act as unquantifiable extra spin-scattering centers and thus cause complicated and indefinable impact on the performance of the spintronic device. Unfortunately, most existing electrode-transfer techniques, either wet or dry processes in electronics, can hardly avoid the above-described risks.

Here in this article, a unique polymer-assistant strain-restricted transfer technique has been developed to guarantee the magnetic property of FM without any change during the transfer process, and meanwhile, construct homogeneous and damage-free FM/NMM interfaces in spintronic devices. In this technique, the strain history of FMs has been restricted at a very low level by a high-modulus polymeric-supporting film. And a huge adhesion difference (about three orders of magnitudes herein) between the polymer films and original or targeted substrates, making the FM electrodes pre-patterned on polymer films can perfectly transfer on targeted substrate covered by channel materials, and then form tight contact. Using this technique, in the first place, FM electrodes, including Co, Ni and $Ni_{80}Fe_{20}$, can be successfully transferred without any change of magnetic, electronic as well as magnetoresistance properties, which has never been achieved before. In the second place, this technique can construct a high-quality interface between FM and channel materials and therefore obtain superior-performance and highly reproducible spintronic devices. In the third place, this technique can successfully prepare spintronic devices based on various spacers, including organic, inorganic, organic-inorganic hybrid, and carbon-based materials, and form superior-quality spintronic devices. Furthermore, this technique also shows the potential of being compatible with large-area arrays of spintronic devices and probing spin-transport mechanisms in even discontinuous channel materials, which will greatly benefit the academic and practical research on spintronics in the future.

## Results

### Polymer-assistant strain-restricted FM transfer technique

Magnetic anisotropy as one of the most important properties of FMs can reflect the magnetic characteristics including coercivity ($H_c$) and magnetic domain and etc., which is also very important for the performance of spintronic devices[17,25]. According to the inverse magnetostriction, stress or strain in FMs will change the magnetocrystalline anisotropy and therefore may alter the direction of the magnetization and the magnetic property of FM. As described by $K_e = -3\lambda\sigma/2$ (for FMs with isotropy magnetostriction, where $K_e$ is the magnetoelastic anisotropy energy, $\lambda$ is magnetostriction coefficient and $\sigma$ is stress)[17], the magnetic property of FMs should be very sensitive to the stress, and such change will be directly reflected on the performance of spintronic devices, such as the curve of magnetoresistance (MR) in spin valve (SV)[2,5,6]. Herein, Co film is used as an example to measure its ultimate strain history of magnetic changes. As shown in Fig. 1a, the magnetic hysteresis curves of Co films versus varied strain history (measured as set in Supplementary Fig. 1a) indicate an obvious and irreversible shift in coercivity when the strain history of Co changed from 1.5% to 1.6%, surely exceeds the elastic strain limit of FMs. At low temperatures, this shift will be much more obvious (see Supplementary Fig. 1b). In other FMs, similar changes in the hysteresis curve along with strain can also be observed, such as $Ni_{80}Fe_{20}$ (from 1.6% to 1.8%, see Supplementary Fig. 1c). Combined with the nearly same conductivity of FM samples before and after threshold strain (see Supplementary Fig. 1d, e), it can be concluded such threshold should only be determined by the change of magnetic domains rather than the electric properties or sample damage. Therefore, according to our experiments, we take 1.5% as the referenced threshold strain history ($\varepsilon_{th}$) of magnetic change of FMs. In fact, such a low strain is almost imperceptible (see Supplementary Fig. 1f) and can be realized by a tiny force at the level of ten-mN (e.g., on 0.2-mm-width and 50-nm-thick Co or $Ni_{80}Fe_{20}$ electrodes employed in this study), which is easy to be induced during transfer by local stress concentration of FMs caused by mechanical curving, twisting and so on.

In this study, we propose a strain-restricted technique to nondestructively transfer the FM electrode with the assistance of a polymeric film (Fig. 1b). In this technique, the FM electrodes pre-patterned on a polymer supporting film are dryly transferred onto the targeted substrate and form tight contact with the targeted substrate after the peeling off and lamination processes (detailed description is shown in Methods). In order to restrict the strain during the whole process, the polymer supporting film should possess a high modulus to restrict the deformation, and the adhesion force of the polymer film with the original substrate should be much lower than with the targeted substrate. Herein, polystyrene (PS) film featuring high modulus is employed as the supporting film, and its high transparency is conducive to electrode alignment. PS film is fabricated by a blade-coating technique[26] on a glass substrate, which can practically avoid the coffee-ring effect and therefore achieve homogeneous thickness and mechanical properties. On such a polymeric film, since the FM electrodes covered part will possess higher strength than the pure PS part, the actual strain on the FM-covered part should be less than that of pure PS part ($\varepsilon_{FM/Poly} < \varepsilon_{Poly}$) under the same stress. Moreover, according to our design that $\varepsilon_b < \varepsilon_{th}$ ($\varepsilon_b$ is the elongation at break of PS film), the PS supporting film should be already broken when the actual strain on FMs is higher than $\varepsilon_{th}$ (here is 1.5%). Such a design provides an intuitive judgment for the change of the magnetic property of transferred FMs. The above-described process that keeps $\varepsilon_{FM/poly} < \varepsilon_{poly} \leq \varepsilon_b < \varepsilon_{th}$ is named as strain-restricted technique, which can serve as one of the foundations of the low-strain transfer of FMs.

In order to guarantee $\varepsilon_b < \varepsilon_{th}$, firstly, the measurements regarding $\varepsilon_b$ of the blade-coating-fabricated PS films with various thicknesses are carried out (the tensile strength versus thicknesses is shown in

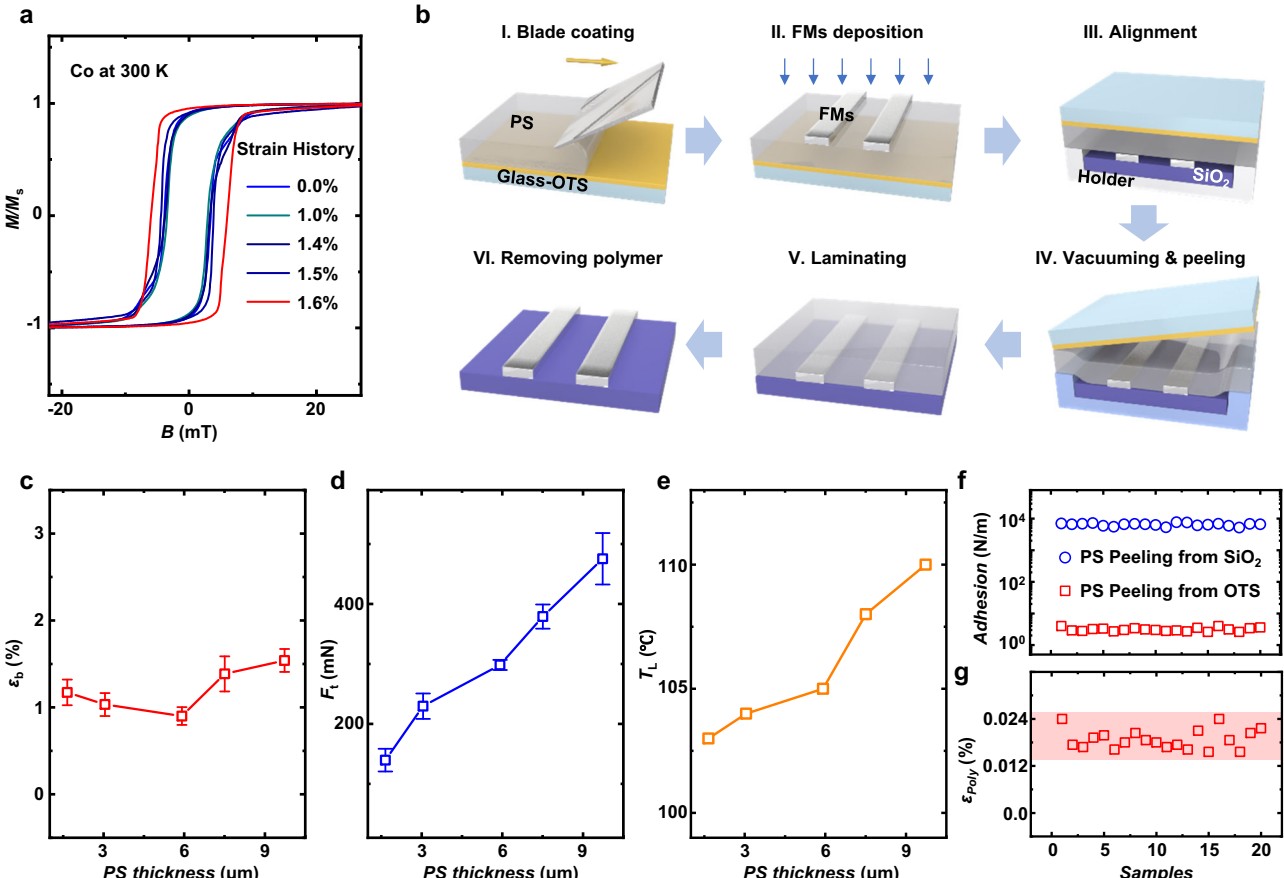

**Fig. 1 | Design of polymer-assistant strain-restricted transfer technique.**
**a** Magnetic hysteresis curves of Co films versus varied strain history. The coercivity of Co shows an obvious shift when strain changes from 1.5% to 1.6%, which means the threshold strain ($\varepsilon_{th}$) of Co is 1.5%. **b** Sketches of the polymer-assistant strain-restricted transfer process for perfectly transferring FM electrodes in a gentle and clean manner. **c**–**e** Measurements regarding (**c**) strain at break ($\varepsilon_b$), **d** tensile force at the strain of 1% ($F_t$), and **e** laminating temperature ($T_L$) of PS versus thicknesses,

which indicates 6-μm PS is a suitable choice. Error bars in **c** and **d** represent the variation range in $\varepsilon_b$ and $F_t$ during the measurements. **f** Comparison of peeling forces of PS films on $SiO_2$ and OTS-modified glass substrates. **g** The suffered strain ($\varepsilon_{poly}$, less than 0.024%) when 6-μm PS peeling off from the glass-OTS substrate, far lower than $\varepsilon_{th}$ of 1.5%. Peeling force and strain as shown in **f** and **g** are measured in 20 randomly selected samples.

Supplementary Fig. 2), in which the lowest $\varepsilon_b$ is around 1% and obtained with the 6-nm-thick PS film. Secondly, tensile force at the lowest strain of 1% ($F_t$) of PS with different thicknesses is measured. As shown in Fig. 1c, d, the PS thickness of 6 μm is finally selected due to the lowest $\varepsilon_b$ along with relatively high $F_t$, which is more beneficial to withstand the effects of peeling force. Moreover, the laminating temperatures of PS films shown in Fig. 1e indicate the lowest temperature for constructing tight FM/substrate contact at various thicknesses, from which the appropriate laminating temperatures of this technique can be obtained.

The other prerequisite for the successful implementation of this technique is to create huge adhesion differences between the employed polymeric films with original or targeted substrates (lower adhesion force between polymeric films with the original substrate than that with the targeted substrate). With the higher adhesion energy between polymeric films and the targeted substrate, the pre-patterned FM electrodes together with polymeric films will be transferred from the original substrate to the targeted substrate homogeneously and gently (Fig. 1b). Such an adhesion difference is caused by controlling the surface energy of substrates. Herein, self-assembly octadecyl trichlorosilane (OTS) has been employed to modify the original substrate (glass) with decreased adhesion energy[27], helping to construct a huge difference in surface energy compared to the targeted substrate. Experimentally, the adhesion differences are quantifically verified by the forces of polymer films peeling off from the

original glass substrate, OTS-modified glass substrate, and target $SiO_2$ substrate. As shown in Fig. 1f and the schematic diagrams in Supplementary Fig. 3, the peeling force between PS and OTS-modified glass is much lower than that between PS and original glass substrate and target substrate, the difference is about three orders of magnitudes, indicating the OTS modification can greatly weaken the adhesive strength between PS and glass. From the transfer process shown in Fig. 1b, it is worth noting that the only risk of the PS film and FM electrode suffering from undesired strain (history) might be in the peeling-off step (diagram IV in Fig. 1b). Our design of this technique can completely address this risk, as shown in Fig. 1g, in which the strain of 6-μm PS can be well controlled merely between 0.015% and 0.025% (far below 1.5%) when peeling off.

**Damage-free transfer proved from FM view**
Here in this manuscript, three commonly used FMs (Co, $Ni_{80}Fe_{20}$, and Ni) have been transferred via the above-proposed technique (as sketched in Fig. 1b). A series of characteristics about FMs before and after transfer were measured, including micro morphologies, magnetic hysteresis curves, electric resistance ratios and anisotropic magnetoresistances (AMR), details are shown in the Methods. As displayed in Fig. 2a, the atomic force microscope (AFM) images show very similar morphology and root-mean-square (RMS) roughness of FMs before and after the transfer, which preliminarily implies the microstructure hasn't been damaged by the mechanical transfer process. Therefore,

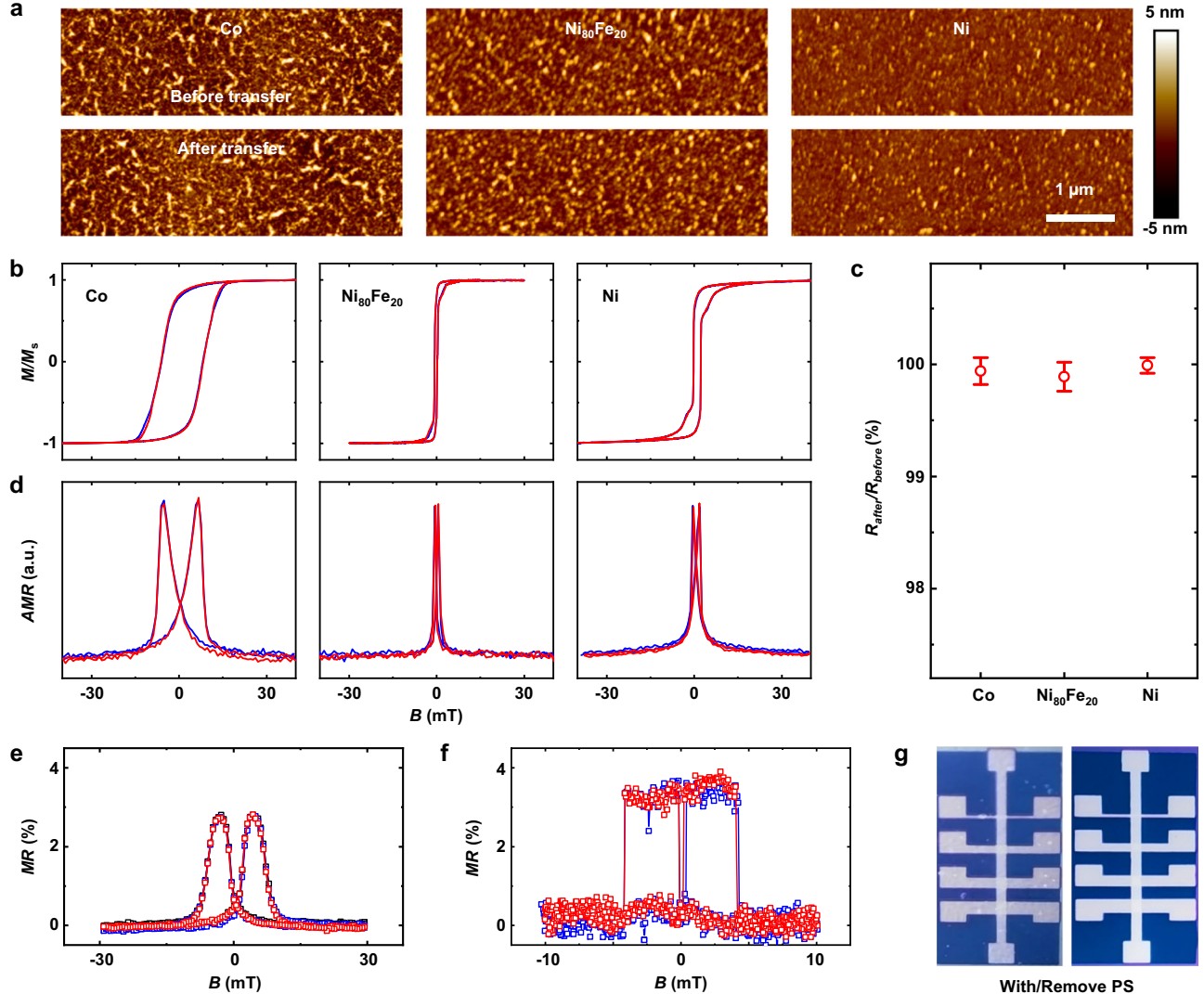

**Fig. 2 | Damage-free transfer of common FM electrodes and tight contact of FM/substrate interface. a–d** Comparison of **a** atomic force-microscope (AFM) images, **b** magnetic hysteresis curves, **c** electrical resistances ratio ($R_{after}/R_{before}$, error bars represent the variation range from 20-electrodes each) and **d** anisotropic magnetoresistance measurements (*AMR*) of Co, $Ni_{80}Fe_{20}$, and Ni before (blue) and after (red) transfer. All the above results suggest nearly no change of FM properties before and after transfer. **e** Magnetoresistance (*MR*) curves of wholly transferred Co/$Al_2O_3$/$Ni_{80}Fe_{20}$ spin valve (SV), before (black) and after (blue) transfer and after removing the PS with solvent (red). **f, g** *MR* curves (**f**) regarding the top-FM-transfer-fabricated Co/$Al_2O_3$/$Ni_{80}Fe_{20}$ device (photos in **g**) before (blue) and after (red) removing PS films. Note that all the *AMR* and *MR* curves in this article are measured with the sweeping magnetic field from negative to positive (corresponding to the right peak of curves) and back to negative (corresponding to the left peak of curves).

all the FMs show almost constant saturation magnetization as well as coercivity according to Fig. 2b, which illustrates that the magnetic properties are preserved perfectly during the transfer process. Besides, the electrical properties of these FM electrodes have been measured and almost constant resistances have been observed before and after the transfer processes (Fig. 2c). Finally, the *AMR* of three FMs electrodes have been measured as shown in Fig. 2d, without change before and after transfer, as expected. All the above results clearly demonstrate that there is almost no change in FM properties before and after transfer.

In fact, the stress on FM electrodes suffered from the peeling process from the original substrate may be anisotropy, however, it is hard to detect in the transferred electrodes with the same-direction alignment (as shown in diagram IV in Fig. 1b). In order to exclude the influence of stress anisotropy on FMs, in the first place, we measure the magnetic property of FMs with the peeling-off direction along and perpendicular with the major axis of FMs, and no change is found. Furthermore, we compare the magnetic-related performance of SVs

with crossed FM electrodes before and after transfer. SV normally structured by two FM electrodes and an NMM media layer has provided the best testbed for this measurement, and its performance (commonly represented by *MR*, detail see Methods) is highly dependent on the magnetic properties of FMs and the interface qualities[5,28]. Herein, SV structured as Co/$Al_2O_3$/$Ni_{80}Fe_{20}$ is fabricated on glass-OTS/PS substrates and then transferred to $SiO_2$ substrates via the proposed technique (see the sketch in Supplementary Fig. 4a). As shown in Fig. 2e, almost constant *MR* curves of SVs before and after transfer have been observed, which implies the magnetic property of FM electrodes aligned in any direction will not be influenced by abovementioned stress anisotropy.

Furthermore, we fabricated the SVs by transferring the top-FM electrodes onto Co/$Al_2O_3$ substrates (see sketch in Supplementary Fig. 4b) and removed the PS film by solvents. The Co/$Al_2O_3$/$Ni_{80}Fe_{20}$ spin valve constructed by the top-FM electrode-transfer technique shows very stable *MR* and macroscopically undamaged structure before and after removing the PS (Fig. 2f, g), which strongly suggests

our transfer technique can lead to a very tight connection between the transferred FM and targeted substrate, even can't be permeated by solvents. By comparing the results as shown in Fig. 2e, f, we find that the peak-like $MR$ curve of wholly transferred SV (Fig. 2e) reveals a slow and gradual switching process between high and low resistance states along with the sweeping magnetic field, which indicates very inconsistent magnetic responses of $Ni_{80}Fe_{20}$ magnetic domains grown on $Al_2O_3$ layer. In comparison, for SVs with transferred $Ni_{80}Fe_{20}$ onto $Al_2O_3$, the $MR$ shows a very sharp switch along with the magnetic field (Fig. 2f), meaning that the $Ni_{80}Fe_{20}$ magnetic domains on PS supporting film possess relatively uniform magnetic response. Such difference can also be reflected in the switching speed of hysteresis loops of $Ni_{80}Fe_{20}$ (on $Al_2O_3$ layer) fabricated by LN-cooling evaporation and transfer technique, respectively (see Supplementary Fig. 5). The $MR$ values and device resistances of SVs with transferred top electrodes also show higher reproducibility as presented via the narrow-distributed superior-quality device performances, in striking contrast to that of SVs prepared by transferring the whole devices (Supplementary Fig. 6).

### Damage-free transfer proved from FM/molecule interface

The above observation provides a foundation from the view of FM magnetic domains for achieving a homogenous and damage-free FM/NMM interface, which will provide a more effective method of constructing spintronic devices based on more fragile channel materials, such as molecular semiconductors. Molecular semiconductors have attracted considerable attention recently from spintronics since they possess very long spin lifetime due to their composition of light elements (such as H, C, N, and O)[29–32]. Because of the fragile essence of molecular semiconductors, the interfacial damage during FM deposition is very serious in molecular spintronic devices[10,11,33,34], which greatly weakens the performance and reproducibility of molecular spintronic devices and limits the rapid development of molecular spintronics. Herein, we construct the SVs based on a molecular semiconductor of [6,6]-phenyl-$C_{71}$-butyric acid methyl ester ($PC_{71}BM$). $PC_{71}BM$ possesses relatively weak spin-orbit coupling (SOC) due to its chemical composition of only carbon and a small number of hydrogen and oxygen, which benefits room-temperature spin transport. Also, the excellent solubility and easy-to-form uniform thin film by simple spin-coating process make $PC_{71}BM$ suitable for constructing SVs by LN-cooling and transferred methods. Herein, we make a direct comparison of device performance prepared by liquid-nitrogen-cooling (LN-cooling) and polymer-assistant strain-restricted transfer technique (Fig. 3a), where the $PC_{71}BM$ layers are 55 nm thick and the bottom electrodes are processed in the same way. It is worth mentioning that the LN-cooling technique (see detailed description in the Methods) should be one of the state-of-the-art technologies for producing high-quality molecular SVs thus far[28,35,36], which is well known for greatly reducing interfacial damage via cryogenic temperature during device fabrication. As shown in Fig. 3b, $MR$ of about 1.2% is observed in an LN-cooling device at room-temperature, and the peak-like $MR$ curve implies the inhomogeneous property of $Ni_{80}Fe_{20}$ magnetic domain on $PC_{71}BM$ spacer, which has been frequently observed in molecule-based SVs[5,37,38]. In contrast, as shown in Fig. 3c, the $PC_{71}BM$-based SV prepared via transfer technique exhibits a significant improvement in $MR$ value. Moreover, the sharp switching between the high and low resistance states suggests that a high-quality and homogeneous interface is formed between the $PC_{71}BM$ and the transferred $Ni_{80}Fe_{20}$ top electrode. The switching magnetic field in the $MR$ curve in Fig. 3c corresponds well to the hysteresis loops of $Ni_{80}Fe_{20}$ after being transferred onto $PC_{71}BM$ channel material provided in Supplementary Fig. 7.

To further reveal the difference of the top $Ni_{80}Fe_{20}/PC_{71}BM$ interfaces in $PC_{71}BM$-based SVs prepared by LN-cooling and transfer methods, respectively, the modified Jullière formula[5] commonly used in molecular spintronics is utilized to evaluate the thickness of the ill-

defined layer (represented the degree of interface damage). By fitting the relationship between the measured $MR$ and the thickness of the molecular spacer, $d$, according to the modified Jullière formula, parameters reflecting the spin-related performance can be obtained. The modified Jullière formula is as follows:

$$MR = \frac{\Delta R}{R_p} = \frac{2P_1P_2e^{-(d-d_0)/\lambda s}}{1 - P_1P_2e^{-(d-d_0)/\lambda s}} \tag{1}$$

Where $\Delta R$ is the difference of device resistances in antiparallel ($R_{ap}$) and parallel ($R_p$) magnetic alignments of FMs, $d_0$ is the average thickness of the ill-defined interfacial layer; $\lambda_s$ is the spin diffusion length of the molecular semiconductors; $P_1$ and $P_2$ represents the spin polarization of injected carriers from FM electrode through the interface and then to the molecular layer. As shown in Fig. 3d, we fit the experimental results of $MR$ at various $d$ in SVs prepared via the LN-cooling and transfer methods using formula (1), the corresponding raw $MR$ curves are shown in Supplementary Fig. 8. The fitted $d_0$ in LN-cooling and transfer-prepared devices are 4.20 nm and 0.01 nm, respectively, implying there is almost no interfacial penetration in the transfer-prepared device, as expected. The fitted $P_1P_2$ in LN-cooling and transfer methods are 0.03 and 0.05, respectively, indicating a more efficient spin injection or detection in the latter devices.

To intuitively observe the interfacial differences, transmission electron microscopy (TEM) has been employed to observe the cross-section of $Ni_{80}Fe_{20}/PC_{71}BM$ interfaces fabricated via different techniques (Fig. 3e, f, the corresponding high-resolution TEM is shown in Supplementary Fig. 9), from which a relatively rough interface can be seen in the LN-cooling sample. Attenuation structure tomography through in situ ultraviolet-visible analysis (Supplementary Fig. 10 and Supplementary Note 1) further qualitatively indicates that the deposited $Ni_{80}Fe_{20}$ can diffuse into the $PC_{71}BM$ layer even at a liquid-nitrogen temperature and therefore form a relatively rough FM/NMM interface. According to the previous literatures[39], the rough profile of FM electrode can induce enhanced local magnetostatic fields, and therefore accelerate spin relaxation in $PC_{71}BM$. Also, the FM atoms diffused into $PC_{71}BM$ will surely enhance the interaction between them, where $Ni_{80}Fe_{20}$ will transfer electrons to $PC_{71}BM$ to change the electronic structure of $PC_{71}BM$ (demonstrated by density functional theory calculation in Supplementary Fig. 11). The diffused nickel-iron ions (not ferromagnetic nickel-iron)[40,41] could lead to an obviously enhanced SOC effect in $PC_{71}BM$ layer near the interface and thus enhance spin relaxation because the SOC strength is directly proportional to $Z^4$ ($Z$ is the atomic number)[42], the detailed explanation can be seen in Supplementary Note 2. From the above points, the spin relaxation in the molecular layer will be intensified in LN-cooling-prepared $PC_{71}BM$-based SVs compared to transfer-prepared devices. In fact, the aggravation of spin relaxation in the spin-injection and spin-transport process can finally reflected in the detected decreased $P_1P_2$ and $\lambda_s$[39]. The obtained $\lambda_s$ according to Fig. 3d have clearly demonstrated the above point, where a 30.9-nm $\lambda_s$ observed in the LT-cooling-prepared device is far less than the $\lambda_s$ (58.8 nm) in the transfer-prepared device.

### Reproducibility and broad application of FM transfer

The reproducibility of the performances of $PC_{71}BM$-based SVs prepared by the transfer technique has also been investigated from the views of both $MR$ and area resistances ($R_A$). As shown in Fig. 4a–c, it clearly shows that obviously narrowed divergence of $MR$ measured from transfer-prepared $PC_{71}BM$-based SVs (4.09 ± 0.17%) compared to that measured from LN-cooling-prepared devices (1.23 ± 0.32%). The high reproducibility of $MR$ of SVs prepared by the transfer technique reveals the effectiveness of this strain-restricted strategy, which guarantees the damage-free magnetic property of FMs and the controllable FM/NMM interface during every transfer process. Moreover, the comparison of $R_A/R_{A-Ref}$ values distribution based on SVs prepared

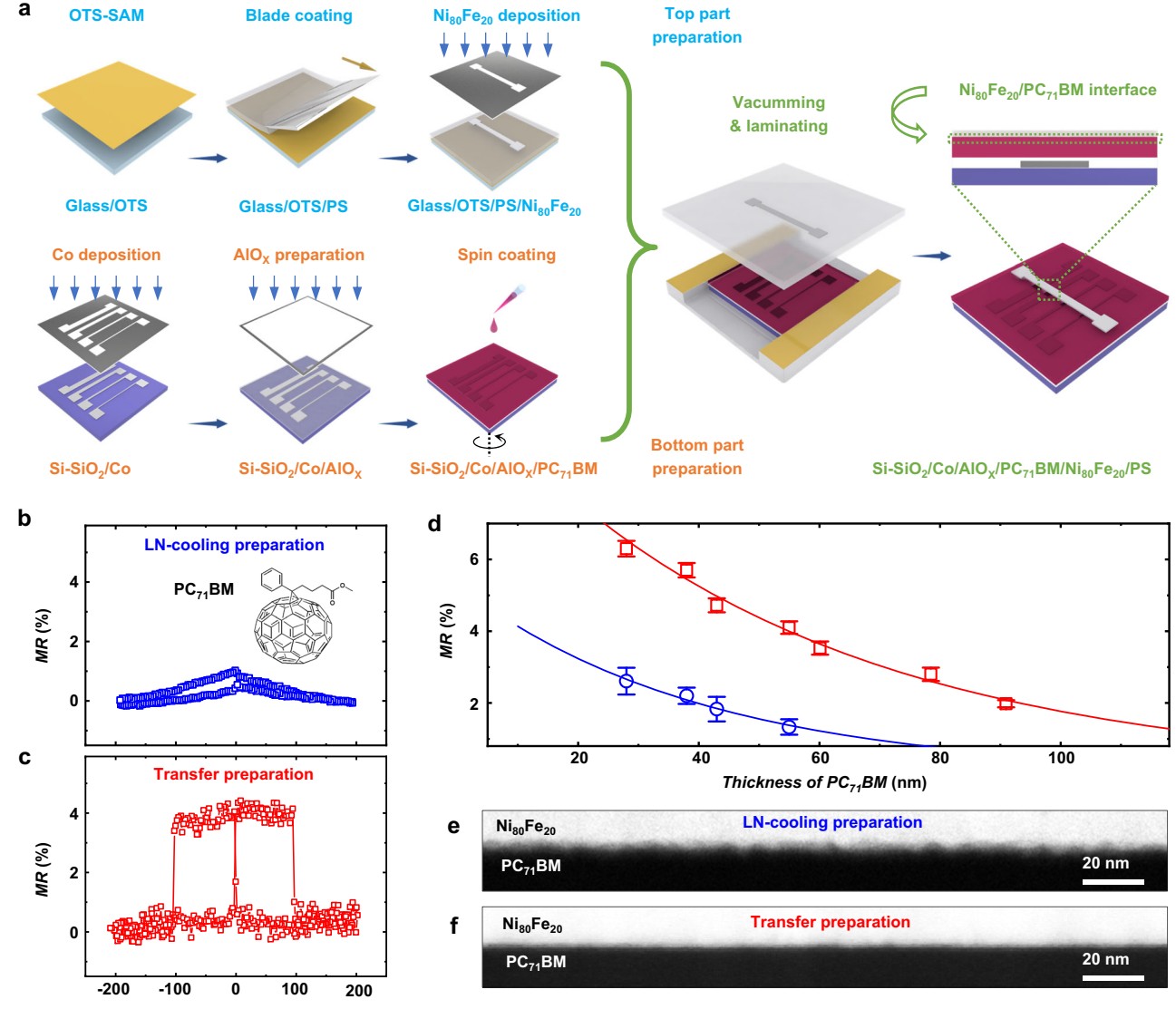

**Fig. 3 | Damage-free FM transfer pursuing superior quality of FM/molecule interface. a** Sketch of the processes for fabricating molecular SV via transfer technique. **b, c** *MR* curves measured in 55-nm-PC$_{71}$BM-based SVs prepared by **b** LN-cooling and **c** transfer techniques. The chemical structure of PC$_{71}$BM is shown in the inset of **b**. **d** Fitting of *MR* data at various thicknesses of PC$_{71}$BM in SVs prepared via LN-cooling (blue square and line) and transfer (red square and line) methods. Error bars indicate the data are collected from 5 devices on the same chip at each thickness. **e, f** Cross-section transmission electron microscopy (TEM) images of Ni$_{80}$Fe$_{20}$/PC$_{71}$BM interfaces prepared by **e** LN-cooling and **f** transfer methods.

by different techniques, as shown in Fig. 4d–f, demonstrates a great improvement in the uniformity of electrical properties in transfer-prepared SVs, in which the data are extracted from 10 devices with diverse thicknesses of PC$_{71}$BM layer. The provided data of SVs with diverse PC$_{71}$BM thickness indicates such result is general and independent of the thickness of the spacer. These observations suggest that the transfer technique should be very helpful in obtaining superior-quality FM/NMM interfaces and repeatable performances in molecular SVs, which must be very important for both academic and practical studies in molecular spintronics in the future. Moreover, due to the uniform high-quality interface (Supplementary Fig. 12) as well as the PS-layer encapsulation in transfer-prepared devices, the stability of spin-valve performance, that is the *MR*, is highly improved even in an air atmosphere (almost no attenuation during 70 days, see Fig. 4g).

Besides the spacer of PC$_{71}$BM whose roughness is very low, the proposed polymer-assistant strain-restricted transfer technique has also been applied for fabricating spintronic devices based on very various kinds of channel materials (Fig. 5). Herein, the employed

channel materials include inorganic (e.g., LiF), organic (e.g., P3HT, CuPc, and Tips-pentacene), organic-inorganic hybrid (e.g., Au$_{25}$ nanoclusters (Au$_{25}$-NCs), see structure in Supplementary Fig. 13a) and carbon-based (covalent organic frameworks (COF), as shown in Supplementary Fig. 13b) materials. The channel materials can be very smooth (*RMS* roughness below 1 nm) or rough (*RMS* roughness nearly 40 nm) and can be continuous or even discontinuous (Fig. 5a), also the channel materials can be prepared in diverse ways (e.g., spin coating, vacuum deposition, blade coating as well as in situ solution growth).

For the smooth and continuous thin films, such as Au$_{25}$-NCs, LiF, COF (spin-coated), and P3HT, it is clear that the *MR* curves measured in transfer-prepared devices always show square-like shapes and higher values of *MR* in contrast to the LN-cooling produced ones (Fig. 5b, c). As analyzed in the above section, these observed *MR* improvements may be induced by the reduced interface damage and optimized device quality, which provides a good channel for spin-dependent carrier transport with a weakened scattering. For the very rough or even discontinuous channel materials, such as CuPc (polycrystalline

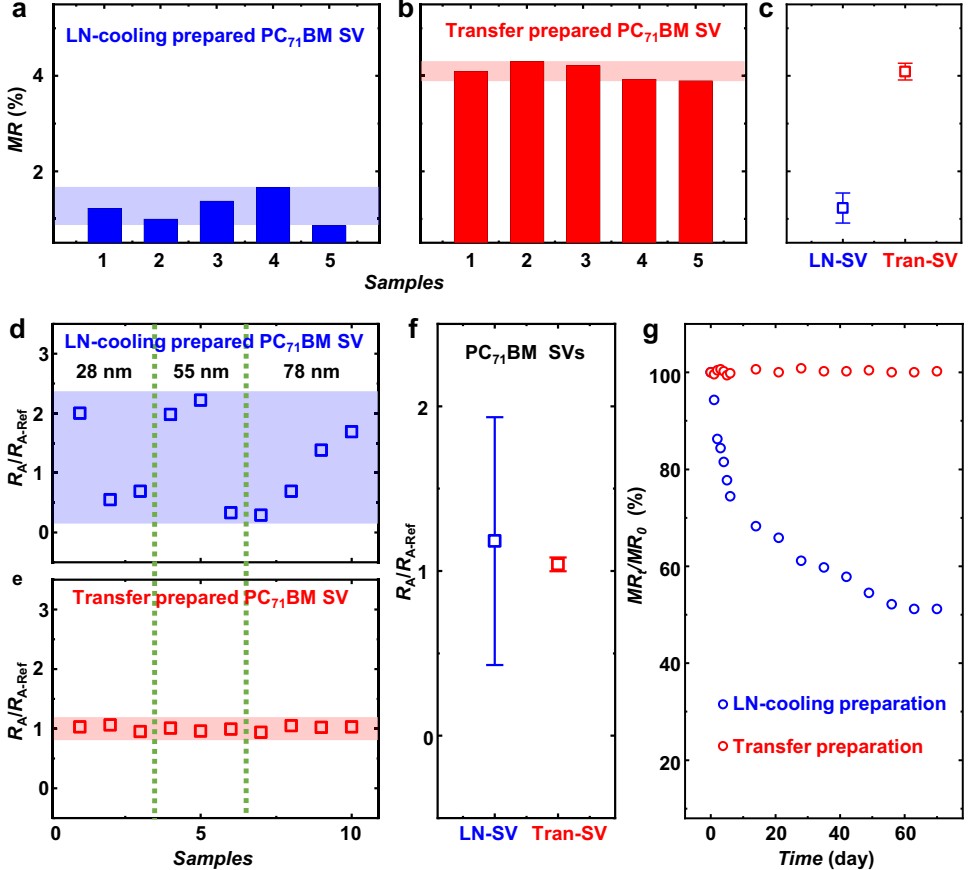

**Fig. 4 | Reproducibility and stability of PC₇₁BM-based SVs prepared by LN-cooling and transfer techniques.** **a**, **b** *MR* data of PC₇₁BM-based SVs in the same chip, prepared by **a** LN-cooling (blue) and **b** transfer (red) techniques, respectively, in which the thickness of PC₇₁BM is 55 nm. **c** The standard deviation of *MR* shown in **a** and **b**, error bars represent the variation range of *MR* from 5 samples. **d**, **e** $R_A/R_{A\text{-Ref}}$ values of 10 SVs based on diverse-thickness PC₇₁BM (28 nm, 55 nm, 78 nm), prepared by **d** LN-cooling and **e** transfer techniques. $R_{A\text{-Ref}}$ is the mean value of $R_A$ of SVs based on the respective PC₇₁BM thickness. $R_{A\text{-Ref}}$ of LN-cooling SV with 28-nm,

55-nm, 78-nm thick PC₇₁BM is 3.42 Ωcm², 31.5 Ωcm², 462 Ωcm², respectively, while the $R_{A\text{-Ref}}$ of transferred SV with 28-nm, 55-nm, 78-nm thick PC₇₁BM is 5.36 Ωcm², 41.7 Ωcm², 454 Ωcm², respectively. The device areas are varied from 100 × 100 μm² to 200 × 500 μm². **f** The standard deviation of $R_A/R_{A\text{-Ref}}$ achieved from 10 samples shown in **d**, **e**. It is obvious that the transfer technique greatly narrows the divergence of resistances of PC₇₁BM-based SVs. **g** Air stability of PC₇₁BM-based SVs prepared via LN-cooling (blue) and transfer (PS reserved) methods (red). $MR_t/MR_0$ refers to the percentage of retentive *MR* over time.

film), Tips-pentacene (single crystal clusters), and COF (via in situ solution growth), the traditional-depositing fabrication, surely including LN-cooling technique, cannot produce qualified devices because these molecular films will lead to short circuit in devices (three diagrams on the right of Fig. 5b). In great contrast, the proposed transfer technique can creatively construct high-quality SVs based on these channel materials (three diagrams on the right of Fig. 5c), the hysteresis loops of transferred Ni₈₀Fe₂₀ onto the above channel materials are provided in Supplementary Fig. 14. The transfer technique actually provides an unprecedented approach for revealing spin-related mechanisms in these materials and exploring their potential applications in future. Moreover, this method can easily accommodate large-scale fabrication (Supplementary Fig. 15), which further illustrates the great potential of this technique.

## Discussion

In summary, we show a general polymer-assistant strain-restricted transfer technique to construct homogenous and damage-free FM/NMM interfaces and therefore lead to superior-quality spintronic devices with high reproducibility. The implementation of this technique is very simple and has no specific requirements on channel materials, which has been demonstrated applicable to universal channel materials in various types and morphologies, including those even discontinuous. In terms of application, this method allows a very

repeatable process of device fabrication and shows very stable device performances due to encapsulation by polymeric films. Encouragingly, preliminary results suggest the potential for large-scale preparation of spintronic devices through this method. In a word, the polymer-assistant strain-restricted transfer technique possesses the advantages of being non-destructive, easy to control, highly reproducible, and widely applicable, which has great potential in reliably probing spin-transport mechanism in various channel materials and therefore promotes the technology transition of spintronics field.

## Methods
### Device preparation
**Strain-restricted transfer technique.** The Ni₈₀Fe₂₀ electrodes were evaporated on a glass/OTS/polymer substrate (1 × 1 cm²), where OTS is self-assembled onto a glass substrate in a vacuum oven at 120 °C and the polymer used in our paper was PS. The top glass/OTS/PS/Ni₈₀Fe₂₀ part was transferred onto the bottom Si-SiO₂ substrate or Si-SiO₂/Co/spacer substrate under nitrogen protection (see Figs. 1b and 3a). Subsequently, due to the huge adhesion difference of polymer between the original and targeted substrate, the glass substrate can be easily peeled off from PS without damaging PS thin film; followed by a vacuum and heat treatment above 106 °C, PS thin film was softened to make sure the Ni₈₀Fe₂₀ electrode closely attached to the targeted substrate. For a flexible array based on the polyethylene terephthalate

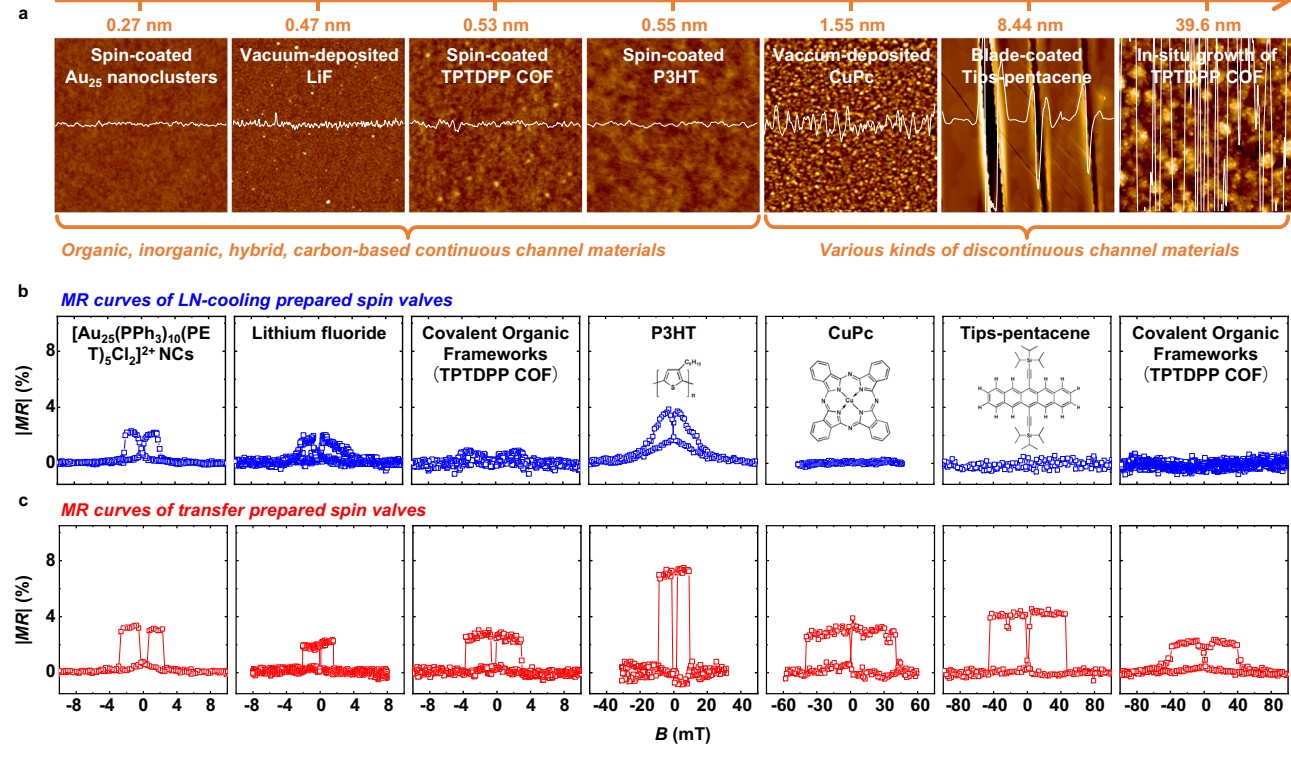

**Fig. 5 | Polymer-assistant strain-restricted transfer technique applied on various channel materials. a** Morphology of various channel materials in spintronic devices, including inorganic (LiF), organic (P3HT, CuPc, Tips-pentacene), organic-inorganic hybrid (Au$_{25}$ nanoclusters), and carbon-based (COFs) materials, which are deposited via diverse methods. The insets are the height amplitude of the section and the height scale (same as the height of the atomic force micrograph) is from −20 nm to 20 nm. **b, c** MR curves measured from SVs based on various channel materials produced by **b** LN-cooling and **c** transfer techniques, respectively.

(PET) substrate (4 × 4 cm², see Supplementary Fig. 15), a 1.5-nm-thick LiF was pre-deposited to modify the PET surface and thus enhance its adhesion with the polymer. Then, by implementing the same deposition and transfer steps as above described with the heat-treatment temperature increased up to 120 °C, the flexible array structured by PET/LiF/Co/spacer/Ni$_{80}$Fe$_{20}$ can be achieved. Also, the PS film can be replaced by PVA, PPO, and PSF in this technique, and the parameters for carrying out the transfer process should surely be changed with that variation. Note, that the successful employment of this technique definitely needs high-intensity practice since we have not converted it into equipment currently. Moreover, the glass/OTS/PS/Ni$_{80}$Fe$_{20}$ part can be prepared in a low-cost and simple way and the glass can be reused by remodifying with OTS, which makes it still potential in cost saving and large-scale preparation.

**Preparation of bottom electrode and spacers in spin valves.** The SVs in this manuscript were constructed with a sandwich structure, where a non-magnetic spacer was sandwiched with the two FM electrodes (Co and Ni$_{80}$Fe$_{20}$) and the device areas were varied from 100 × 100 μm² to 200 × 500 μm². For the preparation of SVs structured as Co/spacers/Ni$_{80}$Fe$_{20}$ (the spacer refers to Al$_2$O$_3$, PC$_{71}$BM, Au$_{25}$ nanoclusters, LiF, P3HT, CuPc, Tips-pentacene, and COF in this study) from bottom to top on Si-SiO$_2$ substrate, eight 20-nm-thick Co electrodes were first deposited with a shadow mask, the evaporating rate was 1 Å s⁻¹. The preparation details of the above spacer layers were as follows: Al$_2$O$_3$ was deposited using an e-beam evaporator. PC$_{71}$BM, P3HT, and Au$_{25}$ nanoclusters were formed by a spin-coating process at 3000 rpm in a nitrogen-atmosphere glove box, where the thicknesses of the molecular layers were controlled by different concentrations. The PC$_{71}$BM and P3HT were dissolved in chloroform, and Au$_{25}$ nanoclusters were dissolved in 1, 2-dichloroethane. LiF and CuPc were thermally evaporated, and their thicknesses were controlled by two film-thickness monitors. COF film was deposited by spin coating (small molecular weight) or in situ solution growth (large molecular weight). And AlO$_x$ interfacial layer between the Co and molecular layer was prepared by depositing an Al layer using an e-beam evaporator and then slightly oxidized by plasma treatment. The thicknesses of spacers were corrected by Ellipsometer (SE-VE-L, Wuhan Eoptics Technology Co., Ltd) and AFM (Bruker Co., Ltd). The prepared bottom electrodes and spacers in SVs were ready for subsequent top-FM electrode transfer or deposition. Note, that all the spacers employed herein have been processed after the same temperature and time as the lamination in the transfer technique.

**Liquid-nitrogen-cooling (LN-cooling) technique for preparing top electrode.** The top Ni$_{80}$Fe$_{20}$ electrodes were directly evaporated onto the spacers by cooling the sample at liquid-nitrogen temperature, Ni$_{80}$Fe$_{20}$ electrodes were gently evaporated at a rate of 0.1 Å/s for the first 2 nm and 1 Å/s for the last 48 nm. It has been demonstrated previously that a combination of rate-control evaporation and liquid-nitrogen cooling of the substrate can largely minimize the damage from the energetic metal atoms.

## Characterization
**Mechanical characterizations.** Strain history of ferromagnetic metals (FMs), stress-strain curves of PS film, and peeling forces of PS films (perpendicular direction) on SiO$_2$ and glass-OTS substrates were measured by Agilent T150, Agilent Technologies, Inc. For the measurement of the strain history of FMs, FMs were deposited on 200 μm-thick PET substrates and then subjected to a constant strain tensile test. After reaching the target strain (such as 1.0%, 1.4%, 1.5%, 1.6%, 1.8%, and 2.0%), the samples were immediately unloaded, transferred, and

installed in the physical property measurement system (PPMS-9, Quantum Design Inc.) for hysteresis loop measurement by vibrating sample magnetometer (VSM). For the measurement of peel-off force, the classical Kendall's model has been employed to do the mechanical analysis in this study, which gives the relation among peel-off force, peeling angle, elastic deformation, and interfacial adhesion energy for an elastic thin-film peeling from a rigid substrate:

$$\Gamma = \frac{F}{\omega}(1 - \cos\theta) + \left(\frac{F}{\omega}\right)^2 \frac{1}{2Eh} \qquad (2)$$

where $\Gamma$ is the interfacial adhesion energy, $F$ is the peel-off force, $\theta$ is the peeling angle, $\omega$, $E$ and $h$ are the width, Young's modulus and thickness of the film, respectively.

**Electrical and magnetic characterizations.** The hysteresis loops of FMs were measured at 300 K and 80 K via a direct current magnetic property measurement by VSM of PPMS. Electrical and magnetic characterizations including resistance and anisotropic magnetoresistance (AMR) of the ferromagnetic electrodes, and magnetoresistance (MR) of all the SVs were carried out by Keithley 4200 Semiconductor Analyzer connected with a low-temperature, high-vacuum, magnetic field probe station (CRX-EM-HF, Lake Shore Co., Ltd.). Typical 4-probe measurement has been employed to avoid contact problems in electrical and magnetic characterizations. By applying a constant bias and sweeping magnetic field on a spin valve, device resistance will be varied with the magnetic field, and the MR values can be calculated by formula (1).

**Atomic force microscopy (AFM) characterization.** The AFM images were measured under tapping mode with a Multimode-8 atomic force microscope produced by Bruker Co., Ltd, and the RMS roughness was also been calculated by the software also provided by Bruker Co., Ltd.

**Cross-section transmission electron microscopy (TEM) characterization.** The cross-section of the SV was prepared by the standard focused ion beam method (Ion-Electron Dual Beam system, FEI Strata DB 235), where 100 nm Al was deposited on the PS surface to protect the surface during cutting. The cross-section TEM images were obtained using an FEI G2 F30 operating at 300 kV.

## Data availability
The data generated in this study are provided within the "Source Data" file. Source data are provided in this paper.

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

## Acknowledgements

This work is supported financially by the National Natural Science Foundation of China (Grant Nos. 52250008, 52050171, 22175047, 51973043, X.S.; 52103203, L.G.; 52303225, X.G.; 52103338, R.Z.), the Strategic Priority Research Program of the Chinese Academy of Sciences (Grant Nos. XDB36020000, XDB0520000, X.S.), the CAS Instrument Development Project (Grant No. YJKYYQ20170037, X.S.), the Beijing Natural Science Foundation (Grant Nos. 4222087, 2222086, X.S.), Natural Science Foundation of Shandong Province (Grant No. ZR2020ME070, X.S.), the State-Funded Postdoctoral Fellows Program of China (Grant No. GZC20230635, Y.Q.), and the CAS Pioneer Hundred Talents Program (X.S.).

## Author contributions

X.S. designed and supervised the work; L.G. co-designed this work and carried out the mechanical testing and part of sample preparation; X.G. organized the whole experiments and carried out the fabrication and characterizations of the electrical and magnetoresistance properties of molecular SVs; S.H. carried out the characterizations of morphological, magnetic properties of FMs, and the fabrication and characterization of inorganic SVs; L.G., X.G., and S.H. contributed equally to this work; W.S. contributed to the early-stage explorations of this work; R.Z. drew most of the sketches; R.Z., J.W., P.M., and K.Z. helped with the TEM characterization; Yayun L. and L.L. helped with the mechanical measurements; G.W. helped with the peeling force analysis and calculation; C.Z. and K.M. helped with the Jullière formula fitting; Cheng Z., Y.Q., A.G., and K.M. helped with device fabrication with various channel materials; K.M. and T.Y. helped with the AFM experiments; X.L. and W.Q. helped in theoretical support and carried out the measurement of attenuation structure tomography through in situ ultraviolet-visible tomography analysis; A.G. and X.Y. helped with contact-angel measurements; Yaling L. helped with the fabrication of COF thin film; L.J. helped with growth and transfer of the Tip-pentacene single crystals. L.G. and X.S. co-wrote this paper with input from the other authors; L.G., X.G., S.H., R.Z., Y.Q., K.M., K.W., W.M., Chuang Z., W.Q., W.Y., and X.S. have worked on processing and presenting the data; all the authors joined the multi-round discussions and revisions of this manuscript.

## Competing interests

The authors declare no competing interests.
