## [Peer Review File · Nature Communications]

REVIEWER COMMENTS

Reviewer #1 (Remarks to the Author):

In this study, the authors have demonstrated a new technique to transfer a metal film to a substrate covered with various materials and its applicability to spintronic devices. Not only single magnetic metal films but also spin-valve junctions can be fabricated using this technique. The transferred device shows a better interfacial quality, magneto-resistance property, and endurance, compared with those fabricated by a conventional deposition technique. The concept is interesting and the experimental data are systematic. This work may open a new route for fabricating spintronic devices with more complicated structures, and provide significant impact on both fundamental and application studies in this field. I can thus recommend the publication of this work in Nature Communications if the following issues are addressed and the manuscript is revised appropriately.

1) Although the authors conducted various measurements using many samples, the experimental details are omitted in the main text, e.g., the device structure, size, sweep direction of magnetic fields. Since this paper claims the reliability in the proposed device fabrication, the measurement methods need to be carefully stated.

2) On Line 91, the authors describe the magneto-elastic coupling energy as $-3*\lambda*\sigma/2$. However, this expression is established only for isotropic magnetostrictive materials. The authors should change the equation to a more general formula or add the explanation that the present equation assumes an isotropic magnetostriction constant.

3) On Lines 95-101, the durability check was performed using the Co or Ni₈₀Fe₂₀ single layer. Although the authors describe the high λ of Co on Line 96, I have a suspicion that there is no relationship between λ and magnetic yielding strain. Actually, the similar changes were observed in the strain-sensitive Co and strain-insensitive Ni₈₀Fe₂₀ ($\lambda \sim 0$). In addition, I do not understand the measurement procedures. After applying a strain, were the samples removed once and installed in PPMS? I cannot find a description of whether the hysteresis loops were determined by Hall or magnetization measurements. The experimental information should be described clearly.

4) Related to 3), the authors should check the electrical resistivity/conductivity change to verify the origin of the threshold. Is the threshold due to the change in magnetic domains or sample damage?

5) In Fig. 2d, why are the AMR curves asymmetric?

6) Why is a data point for the sample with a 20-nm-thick spacer missing in Fig. 3d, while the sample was used in Fig. 4e? The larger TMR ratio should be observed in the sample with a thinner spacer layer. To clarify the reliability of the transfer technique, the data point and raw MR curves should be shown.

7) In Figs. 4d and 4e, the ratio between RA and averaged RA is shown. Shouldn't the vertical axis be written with a 100% order? I cannot understand why the ratio is only $\sim 1\%$. In addition, the author should show the absolute values of RA with the actual device size.

8) Supplementary Fig. 8 is confusing because the substrate to which the metallic films are transferred in this study is the Si/SiO_x substrate, not flexible. Please add the details.

Reviewer #2 (Remarks to the Author):

The manuscript reported a polymer-assistant strain-restricted transfer technique, which can perfectly transfer the pre-patterned ferromagnetic electrodes onto channel materials without any damage or change in the magnetic properties. The results demonstrated that this transfer technique can produce a high-quality interface between ferromagnetic metal and molecule film, thus, magnetoresistance can be significantly enhanced. The sample preparation and metal transfer technique presented in the manuscript is at a very high level. However, the origin of magnetoresistance improvement in spin valves is not clear. The detailed transfer process is not presented. The data quality looks poor and insufficient to support the conclusions. In addition, the hysteresis loops of the transfer-fabricated spin valves are not provided. At present, I can't support publication on Nature Communication. I ask the authors to give detailed answers to my questions on the experimental aspect:

1. Compared with the van der Waals (vdW) magnetic tunnel junctions, what is the interface quality difference between the vdW interface and the strain-restricted-transferred interface? Meanwhile, please give a detailed interpretation of the adhesion difference between vdW interface and the transfer-fabricated interface.
2. In line 74, "And a huge adhesion difference between the polymer films and original or targeted substrates..." Please give a quantity analysis of the "huge adhesion difference" and how to control or adjust this "huge adhesion difference".
3. The magnetoresistance results difference in Fig.2e and 2f originates from the inconsistent or uniform magnetic responses of Ni₈₀Fe₂₀, please validate this conclusion with hysteresis loops or magnetic domain measurements.
4. Please introduce PC71BM to the ordinary reader and reason for choosing this material.
5. Please check equation (1) to ensure its correctness. And provide the parameters (P1, P2) values.
6. As we can see from Fig.3d, why not keep the same thickness of PC71BM in LN-cooling or transfer preparation spin valves when fitting equation (1)?
7. Attenuation structure tomography in supplementary Fig.5 needs more explanation and description. Moreover, the attenuation structure tomography of the transferred Ni₈₀Fe₂₀/PC71BM interface should also be provided.
8. Supplementary Fig.7 should be revised for clarity. Please add a scale bar in supplementary Fig.8.
9. Was any adhesion analysis performed on the transferred Ni₈₀Fe₂₀/PC71BM? A major point that should be addressed is the generality of the polymer-assistant strain-restricted transfer technique. The readers are curious about the generality of this novel transfer technique. As for the transferred metal on various-kind channel materials, there is a lot of characterization around the magnetoresistance, but no accordingly hysteresis loops or adhesion analysis.

The submitted article deals with the formation of strain-restricted ferromagnetic electrodes, which are essential to high quality spintronic devices. A major disadvantage of the energetic metal deposition process is the possibility of the introduction of defects, disorder, etc. into non-magnetic material, resulting in a partial degradation of device performance and a lack of reproducibility. The proposed polymer-assistant strain-restricted transfer technique provides an alternative method to address the issues related to the metal deposition process, in which the electrodes are transferred onto the material of interest. The topic is rather interesting and may attract much interest from the scientists working in the field of spintronics. The experiments and results appear to substantiate the conclusions and interpretations in the submission. Prior to the acceptance of submission, some minor revisions listed below need to be considered in earnest:

1. In Figure 1a, as the stress increases from 1.5% to 1.6%, the area of the loop increases significantly. Is the material exhibiting plastic deformation, as opposed to just elastic deformation? Maybe the sudden change in the hysteresis loop indicates that the material is approaching its yield point.
2. Are there any wrinkles formed during the transfer process? I think heat transfer processes can cause wrinkles in the material being transferred due to the expansion and contraction of the material. Is there a stress impact associated with these wrinkles?
3. Is this transfer process repeatable? How many times can the glass-OTS/PS substrates sustain for such process including metal deposition and transfer? In the previous study (Nature volume 544, pages340–343(2017)), graphene/Ge substrate can be re-used several times for the transfer of various metal electrodes. The author should exhibit the potential of the proposed polymer-assistant strain-restricted transfer technique.
4. Low quality images are shown in Figures 3 e, f. The author should provide more evidence to support the statement, such as high-quality cross-sectional HRTEM image for interface.

Point-by-Point Response to Reviewer's Comments

Response to Reviewer #1:

Question 1-1:

In this study, the authors have demonstrated a new technique to transfer a metal film to a substrate covered with various materials and its applicability to spintronic devices. Not only single magnetic metal films but also spin-valve junctions can be fabricated using this technique. The transferred device shows a better interfacial quality, magneto-resistance property, and endurance, compared with those fabricated by a conventional deposition technique. The concept is interesting and the experimental data are systematic. This work may open a new route for fabricating spintronic devices with more complicated structures, and provide significant impact on both fundamental and application studies in this field. I can thus recommend the publication of this work in Nature Communications if the following issues are addressed and the manuscript is revised appropriately.

Answer:

We appreciate the reviewer's kind comments very much.

Question 1-2:

1) Although the authors conducted various measurements using many samples, the experimental details are omitted in the main text, e.g., the device structure, size, sweep direction of magnetic fields. Since this paper claims the reliability in the proposed device fabrication, the measurement methods need to be carefully stated.

Answer:

Thanks very much for reviewer's useful comments and suggestions. According to reviewer's suggestions, we have added experimental details in the main text and the Methods section. Please see the table below.

Corresponding Revisions Made Due to Question 1-2

Revisions	Responding to
"Note that all the AMR ... (corresponding to the left peak of curves)" has been added in lines 203-206, page 10.	Sweep direction of magnetic fields
"The $\text{Ni}_{80}\text{Fe}_{20}$ electrodes ... can be achieved." has been rewritten in lines 377-388, pages 18-19.	Experimental details, device structure and size
"The SVs in this manuscript ... to $200 \times 500 \mu\text{m}^2$." in lines 395-397, page 19.	Device structure and size
"For the measurement of strain history of FMs, ... by vibrating sample magnetometer (VSM)." has	Measurement method detail of threshold strain

been added in lines 423-427, page 20.	
“The hysteresis loops of FMs ... by VSM of PPMS.” has been rewritten in lines 435-436, page 21.	Measurement method detail of hysteresis loops
“By applying a constant bias ... calculated by formula (1).” has been rewritten in lines 441-442, page 21.	Measurement method detail of MR curves

Question 1-3:

2) On Line 91, the authors describe the magneto-elastic coupling energy as $-3*\lambda*\sigma/2$. However, this expression is established only for isotropic magnetostrictive materials. The authors should change the equation to a more general formula or add the explanation that the present equation assumes an isotropic magnetostriction constant.

Answer:

Thanks very much for reviewer’s kind suggestions. We agree with the reviewer’s comment. In the revised manuscript, we have added the illustration that this equation assumed an isotropic magnetostriction constant. Related revisions have been yellow highlighted in the revised manuscript, shown as “According to the inverse magnetostriction, ... for FMs with isotropy magnetostriction,” in lines 90-93, page 5 of the revised manuscript.

Question 1-4:

3) On Lines 95-101, the durability check was performed using the Co or Ni80Fe20 single layer. Although the authors describe the high λ of Co on Line 96, I have a suspicion that there is no relationship between λ and magnetic yielding strain. Actually, the similar changes were observed in the strain-sensitive Co and strain-insensitive Ni80Fe20 ($\lambda \sim 0$). In addition, I do not understand the measurement procedures. After applying a strain, were the samples removed once and installed in PPMS? I cannot find a description of whether the hysteresis loops were determined by Hall or magnetization measurements. The experimental information should be described clearly.

Answer:

Thanks very much for reviewer’s comments and we agree with the reviewer. In order to avoid misunderstanding the readers, we have deleted the sentence of “since Co with relatively high λ should be more sensitive to strain” in line 97, page 5 of the original manuscript. With regard to the measurement procedure, in fact, after applying a strain, the samples were removed immediately and installed in PPMS, where the hysteresis

loops were measured by vibrating sample magnetometer (VSM). We have added more details in the Methods section of revised manuscript, please see “For the measurement of strain history of FMs, ... by vibrating sample magnetometer (VSM).” in lines 423-427, page 20; “The hysteresis loops of FMs were measured ... by VSM of PPMS.” in lines 435-436, page 21.

Question 1-5:

4) Related to 3), the authors should check the electrical resistivity/conductivity change to verify the origin of the threshold. Is the threshold due to the change in magnetic domains or sample damage?

Answer:

Thanks very much for reviewer’s helpful comments. We have added the conductivity measurements of ferromagnetic electrode samples (including Co and Ni₈₀Fe₂₀) before and after a threshold strain (1.6% and 1.8% respectively), as shown in Supplementary Fig. 1d and 1e. The almost unchanged conductivities imply the origin of threshold strain is not the change of conductivity. Moreover, after applying the threshold strain, the sample is intact from the appearance, implying the threshold cannot be caused by the sample damage. Together with the measurements of hysteresis loops, it can be concluded that the change of magnetic domains causes the strain threshold. Above analysis reveal that the sample was not largely damaged on electrical property when the magnetic property started to change, this is also the reason why we use the change of magnetic property to determine the threshold strain. Accordingly, a sentence has been added as “Combined with the nearly-same conductivity ... or sample damage.” in lines 102-105, page 5 of the revised manuscript.

Supplementary Fig. 1d & 1e

d, Current-voltage curves of Co electrodes before (blue line) and after (red line) a strain of 1.6%. **e**, Current-voltage curves of Ni₈₀Fe₂₀ electrodes before (blue line) and after (red line) a strain of 1.8%.

Question 1-6:

5) In Fig. 2d, why are the AMR curves asymmetric?

Answer:

Thanks very much for reviewer's comments. The previous Fig. 2d only provides the AMR curves of FMs sweeping from negative to positive magnetic fields, thus these curves are asymmetric. We are sorry for confusing the reviewer and readers. In case of misunderstanding readers, the whole AMR curves (with the sweeping magnetic field from negative to positive and back to negative) have been provided as shown in Fig. 2d of the revised manuscript, which obviously are symmetric.

Fig. 2 | d, anisotropic magnetoresistance measurements (AMR) of Co, Ni₈₀Fe₂₀ and Ni before (blue) and after (red) transfer.

Question 1-7:

6) Why is a data point for the sample with a 20-nm-thick spacer missing in Fig. 3d, while the sample was used in Fig. 4e? The larger TMR ratio should be observed in the sample with a thinner spacer layer. To clarify the reliability of the transfer technique, the data point and raw MR curves should be shown.

Answer:

Thanks very much for reviewer's useful comments and suggestions. We are deeply sorry that we have mislabeled the 28 nm (in Fig. 4d and 4e) as 20 nm in previous manuscript, and we are really grateful to the reviewer for pointing out this mistake. In fact, the 20 nm is the experimentally-aimed thickness before fabrication (planned thickness), while 28-nm is the corrected thickness after fabrication in our experiments (actual thickness). We apologize for making such a huge careless mistake. Moreover, all the labeled thicknesses in Fig. 4d and 4e (and their captions) have been rechecked and revised accordingly. Therefore, MR data point of transferred SV with 28-nm-thick spacer, together with additional MR data points of transferred SVs with the same spacer thickness (38 nm, 43 nm) to that of LN-cooling prepared SVs have been added in the revised Fig. 3d, and the raw MR curves of all the data points shown in Fig. 3d have been provided in Supplementary Fig. 8. Accordingly, corresponding contents have been added in the revised manuscript and Supplementary Materials. Please see Fig. 3d in

page 13 of the revised manuscript; “the corresponding raw *MR* curves are shown in Supplementary Fig. 8” in line 265, page 12 of the revised manuscript; Supplementary Fig. 8 and its caption in page 11 of the Supplementary Materials.

Fig. 3 | d, Fitting of *MR* data at various thicknesses of $PC_{71}BM$ in SVs prepared via LN-cooling (blue square and line) and transfer method (red square and line).

Supplementary Fig. 8

a-c, *MR* curves of $PC_{71}BM$ -based SVs prepared by LN-cooling technique, where the thicknesses of $PC_{71}BM$ are 28 nm, 38 nm, 43 nm, respectively. **d-i**, *MR* curves of $PC_{71}BM$ -based SVs prepared by transfer technique, where the thicknesses of $PC_{71}BM$ are 28 nm, 38 nm, 43 nm, 60 nm, 78 nm, 92 nm, respectively.

Question 1-8:

7) In Figs. 4d and 4e, the ratio between R_A and averaged R_A is shown. Shouldn't the vertical axis be written with a 100% order? I cannot understand why the ratio is only ~1%. In addition, the author should show the absolute values of R_A with the actual device size.

Answer:

We appreciate very much for reviewer's comments and feel sorry for mistaking the vertical axis of Figs. 4d and 4e, as well as 4f. And we have corrected their left vertical axis by deleting %, since it is a ratio of resistance and it is not suitable to be expressed as a percentage. In addition, the absolute values of R_A of spin valves with different $PC_{71}BM$ thicknesses and different preparation methods have been added in the captions of Figs. 4d and 4e. And the device size has been clarified in the caption and Methods section, please see " R_{A-Ref} of LN-cooling SV ... from $100 \times 100 \mu m^2$ to $200 \times 500 \mu m^2$." lines 323-326, page 16 of the revised manuscript.

Question 1-9:

8) Supplementary Fig. 8 is confusing because the substrate to which the metallic films are transferred in this study is the Si/SiO_x substrate, not flexible. Please add the details.

Answer:

Thanks very much for reviewer's kind reminder. In Supplementary Fig. 8 of the previous version (now it is Supplementary Fig. 15 in revised version), the substrate used is polyethylene terephthalate (PET). For the transfer procedure, a 1.5-nm-thick LiF is pre-deposited onto the PET substrate in order to enhance the adhesion force between the polymer film and PET, and the lamination temperature should be increased to 120 °C, other steps are same to that based on SiO₂ substrate. We have added more details in Methods section of the revised manuscript and Supplementary Materials. Please see "For a flexible array based on ... can be achieved." in lines 384-388, page 19 of the revised manuscript; "Photo of large-area CuPc-based SV arrays based on flexible PET substrate ($4 \times 4 \text{ cm}^2$) and processed via transfer technique." in the caption of Supplementary Fig. 15, page 18 of the revised Supplementary Materials.

Response to Reviewer #2:

Question 2-1:

The manuscript reported a polymer-assistant strain-restricted transfer technique, which can perfectly transfer the pre-patterned ferromagnetic electrodes onto channel materials without any damage or change in the magnetic properties. The results demonstrated that this transfer technique can produce a high-quality interface between ferromagnetic metal and molecule film, thus, magnetoresistance can be significantly enhanced. The sample preparation and metal transfer technique presented in the manuscript is at a very high level. (I) However, the origin of magnetoresistance improvement in spin valves is not clear. (II) The detailed transfer process is not presented. The data quality looks poor and insufficient to support the conclusions. (III) In addition, the hysteresis loops of the transfer-fabricated spin valves are not provided. At present, I can't support publication on Nature Communication. (IV) I ask the authors to give detailed answers to my questions on the experimental aspect:

Answer:

Thanks very much for reviewer's comments. According to the queries raised by reviewer, we have revised the manuscript and Supplementary Materials accordingly.

(I) In the first place, for the origin of improved magnetoresistance (MR) of spin valves fabricated by the polymer-assistant strain-restricted transfer technique, besides the existing analysis, more demonstrations and descriptions have been added (partial description can also be seen in the answer of Question 2-8): For LN-cooling methods, the FM atoms penetrated into $PC_{71}BM$ will surely enhance the interaction between them (electronic structure will also change), $Ni_{80}Fe_{20}$ will transfer electrons to $PC_{71}BM$ (based on density functional theory simulation, Supplementary Fig. 11) to form nickel-iron ions in $PC_{71}BM$ layers. As a result, nickel-iron ions and $PC_{71}BM$ can be confined together to form a complex through nickel-iron⁺- $PC_{71}BM$ ⁻ interaction. Nickel-iron ions (not ferromagnetic nickel-iron) in $PC_{71}BM$ layer could lead to a pronounced SOC effect in $PC_{71}BM$ layer to enhance spin relaxation. Thus, such large spin relaxation in $PC_{71}BM$ layer originated from LN-cooling depositing $Ni_{80}Fe_{20}$ will weaken the spin injection and transport in $PC_{71}BM$ layer. In contrast, the polymer-assistant strain-restricted transfer technique will effectively prevent the penetration of $Ni_{80}Fe_{20}$ into $PC_{71}BM$ layer, leading to a relatively weak spin relaxation and thus enhanced spin injection and spin transport, finally a larger MR is obtained. Moreover, the analysis of the results of Jullière formula fitting shows the spin injection efficiency (P_1P_2) and spin diffusion length (λ_s) of LN-cooling-prepared samples are lower than that of transferred interface, which can also support above inference. In conclusion, the transfer technique helps to construct interface without FM atoms diffusion, which enables to contribute low spin scattering at interface and thus improve spin transport and device performance. The

corresponding revision please see “Also, the FM atoms diffused into ... seen in Supplementary Note 2.” in lines 287-293, page 14 of the revised manuscript; “spin-injection and” added in line 295 and “ P_1P_2 and” added in line 296; Supplementary Note 2 in page 3 of the Supplementary Materials; References 40, 41 in lines 527-529, page 24 of the revised manuscript.

Supplementary Note 2: Reason for the magnetoresistance improvement of SV prepared by polymer-assistant strain-restricted transfer technique.

The reasons for the magnetoresistance (MR) improvement of SV prepared by polymer-assistant strain-restricted transfer technique has been analysed as follow. For LN-cooling methods, the FM atoms penetrated into $PC_{71}BM$ will surely enhance the interaction between them (electronic structure will also change), $Ni_{80}Fe_{20}$ will transfer electrons to $PC_{71}BM$ (based on density functional theory simulation, Supplementary Fig. 11) to form nickel-iron ions in $PC_{71}BM$ layers.^{1,2} As a result, nickel-iron ions and $PC_{71}BM$ can be confined together to form a complex through nickel-iron⁺- $PC_{71}BM$ ⁻ interaction. Nickel-iron ions (not ferromagnetic nickel-iron) in $PC_{71}BM$ layer could lead to a pronounced SOC effect in $PC_{71}BM$ layer to enhance spin relaxation. Thus, such large spin relaxation in $PC_{71}BM$ layer originated from LN-cooling depositing $Ni_{80}Fe_{20}$ will weaken the spin injection and transport in $PC_{71}BM$ layer. In contrast, the polymer-assistant strain-restricted transfer technique will effectively prevent the penetration of $Ni_{80}Fe_{20}$ into $PC_{71}BM$ layer, leading to a relatively weak spin relaxation and thus enhanced spin injection and spin transport, finally a larger MR is obtained.

Supplementary Fig. 11

a-c, Density functional theory (DFT) calculation models of nickel and iron atoms penetrating into the $PC_{71}BM$ layer from none (**a**) to small (**b**) and large amounts (**c**), the corresponding charge transfer from $Ni_{80}Fe_{20}$ to $PC_{71}BM$ layer are zero, 0.42, 0.58 electrons. **d-f**, Energy band structure of interfacial $PC_{71}BM$ molecules affected by

different degrees of Ni₈₀Fe₂₀ penetration in spin valves from none (representing transferred PC₇₁BM/Ni₈₀Fe₂₀ interface) (**d**) to small (**e**) and large amounts (representing LN-cooling PC₇₁BM/Ni₈₀Fe₂₀ interface) (**f**), where black and red lines represent spin-up and spin-down electrons, respectively.

(II) In the second place, according to the reviewer’s suggestion, more details of the transfer process and measurements have been added. Please see the table below.

Corresponding Revisions Made Due to Question 2-1-II

Revisions	Responding to
“(lower adhesion force ... with the targeted substrate)” has been added in lines 150-151, page 7.	Requirement of transfer technique
“Experimentally, the adhesion differences ... three orders of magnitudes,” has been rewritten in lines 157-162, pages 7-8.	Details of transfer procedure
“The Ni ₈₀ Fe ₂₀ electrodes were evaporated ... can be achieved.” has been rewritten in lines 377-388, pages 18-19.	Details of transfer procedure
Supplementary Fig. 3 has been added.	Details of transfer procedure
“Note that all the AMR ... (corresponding to the left peak of curves).” has been added in lines 203-206, page 10.	Measurement details of AMR
“For the measurement of strain history of FMs, ... by vibrating sample magnetometer (VSM).” has been added in lines 423-427, page 20.	Measurement detail of threshold strain
“The hysteresis loops ... by VSM of PPMS.” has been rewritten in lines 435-436, page 21.	Measurement detail of hysteresis loops

(III) In the third place, according to the reviewer’s useful comments, the hysteresis loops of the transfer-fabricated spin valves have also been measured, as shown in Supplementary Fig. 7 and Fig. 14. The detailed illustration please refer to the response of Question 2-10.

(IV) Finally, to sufficiently support the conclusion that the transferred SVs possess

improved *MR* and the high reproducibility, more data (see the details below) has been added according to reviewer's following comments, and we believe these supplements can reach the high requirement of the reviewer. All related revisions have been yellow highlighted in the revised manuscript and Supplementary Materials.

Question 2-2:

1. Compared with the van der Waals (vdW) magnetic tunnel junctions, what is the interface quality difference between the vdW interface and the strain-restricted-transferred interface? Meanwhile, please give a detailed interpretation of the adhesion difference between vdW interface and the transfer-fabricated interface.

Answer:

Thanks very much for reviewer's comments.

For the first question, we think the reviewer is right to compare vdW interface with strain-restricted transferred FM/molecule interface. In most cases, as there is no chemical bonds between the Ni₈₀Fe₂₀ metal and organic molecules (without considering the cases of spinterface effects), the interaction between metal and molecules is normally in the range of vdW force. Although the Ni₈₀Fe₂₀/molecule interface is not atomically sharp as in vdW heterostructures, the Ni₈₀Fe₂₀ metal does form a conformal coating to the molecular layer as shown by TEM images (see Fig. 3f, as well as new-added high-resolution TEM images in Supplementary Fig. 9). However, we should also state that such transferred interface, formed between diverse spacer materials (except inert ones) and electrodes, may induce more complex interaction that is not limited to vdW contact (e.g. combined effect caused by chemical reactions and physical adsorptions). In most cases, we do not concern the types of interfacial contact (actually, it highly depends on the property of materials themselves, both electrode and spacer, rather than fabrication technique), and surely it is difficult to give a definite quality comparison between transferred interface and the vdW interface. Besides, the target of our strain-restricted-transferred technique is to construct a clean and uniform-contact interface (similar to the electrode transfer on 2D-materials in reference Nat. Electron., 2019, 2, 187-194.), and thus help to form uniform interface interaction and high-efficient charge and spin injection, which is of great importance for building repeatable SVs. In our opinion, the targets of our strain-restricted-transferred interface and vdW interface are not exactly consistent except for clean. From this perspective, a detailed comparison of the two interface qualities is not appropriate in a general sense.

For the second question, for vdW contact (such as inert molecule/FM interface in some cases), because vdW force scales with $1/r^6$, where r is the distance of interaction, the conformal coating of Ni₈₀Fe₂₀ onto molecules minimizes the distance and hence improves the strength of the interaction. However, as illustrated above, the contact types produced by our transfer technique are not limited to vdW type, and comparison of

these two interface properties is not appropriate (including the adhesion strength). Also, to our best knowledge, the exact strength of adhesion can vary with that of vdW heterostructure and can depend on the method of measurement. In conclusion, although the reviewer is right, adhesion is a useful figure of merit for the quality of the interface, it is clear such study forms a subject of its own and goes beyond the focus of this paper.

Question 2-3:

2. In line 74, "*And a huge adhesion difference between the polymer films and original or targeted substrates...*" Please give a quantity analysis of the "*huge adhesion difference*" and how to control or adjust this "*huge adhesion difference*".

Answer:

Thanks very much for reviewer's useful comments.

In the first place, we would like to apologize for using inaccurate word "huge difference" to describe an important process in our study. By specific experimental measurements, we found that an adhesion difference larger than three orders of magnitude between the polymer films and targeted substrates can always be obtained, we describe it as "huge adhesion difference" previously. Accordingly, more precise description of "(about three orders of magnitudes herein)" has been added in lines 73, page 4 of the revised manuscript.

In the second place, the quantity analysis of the "adhesion difference" is conducted by peeling force measurements in this study. As shown in Fig. 1f of the revised manuscript, the peeling force of PS thin film from the original OTS-modified glass substrate and the target SiO₂ substrate have been provided, the schematic diagram is added in Supplementary Fig. 3a and 3b. According to reviewer's comments, the manuscript has been revised to describe the result more precisely, as shown in the yellow-highlighted words "Experimentally, the adhesion differences ... three orders of magnitudes," in lines 157-162, pages 7-8 of the revised manuscript. It should be noted that, for the analysis of the adhesion differences between polymer films and original or targeted substrates, there is a large difference between experiment and theory (such as surface energy measurements and calculations) due to the actual surface may be influenced by the environment. Therefore, after discussing with several professors from mechanics (some of them are coauthors of this work), we have forced to give up describing this process theoretically (due to great inaccuracy), and here we only use the experimental method to give a quantity analysis of the adhesion difference.

In the third place, as for controlling and adjusting the adhesion difference, in our experiment, it requires the adhesion of polymer with targeted substrate should be much higher than that with the original substrate. We found the transferred and thermal laminated PS thin film possess relatively high adhesion force with most of materials (including almost all organic semiconductors, most inorganic materials, and all of the

materials shown in this paper), therefore, the key to enhance adhesion difference is to reduce the adhesion force between PS and the original substrate. Herein this manuscript, OTS is used to modify the original substrate and thus reduce the adhesion force of PS with original substrate even close to negligible. OTS modification on glass substrate is the combination of chemical modification and physical adsorption (as shown in the Supplementary Fig. 3a), where the physical adsorption plays a more important role on reducing the adhesion force. With the different OTS-modified conditions (like OTS dosage, treatment time and temperature) of the original substrate, the adhesion force of PS and the original substrate can be controlled and adjusted.

Supplementary Fig. 3

a, Schematic diagram of peeling off polymer thin film from the OTS-modified original substrate, where the peeling force is relatively small. The OTS modification contains chemical adsorption (linked to the original substrate) and physical adsorption (residual OTS), and the physical adsorption plays a crucial role on reducing the adhesion force between polymer and the original substrate. **b**, Schematic diagram of peeling off polymer thin film from the unmodified original substrate, where the peeling force is large and it is hard to peel off the polymer. **c**, Schematic diagram of peeling off polymer thin film from the targeted substrate in an already laminated sample, where the peeling force is large and it is hard to peel off the polymer.

Question 2-4:

3. The magnetoresistance results difference in Fig.2e and 2f originates from the inconsistent or uniform magnetic responses of Ni₈₀Fe₂₀, please validate this conclusion with hysteresis loops or magnetic domain measurements.

Answer:

Thanks very much for reviewer's comments. According to the reviewer's suggestions, the hysteresis loops of LN-cooling deposited and transferred Ni₈₀Fe₂₀ on PC₇₁BM spacers have been measured and added in Supplementary Fig. 5 in the revised version. As shown in Supplementary Fig. 5, the variation of magnetic moment with applied sweeping magnetic field in transferred Ni₈₀Fe₂₀ is obviously sharper than that of LN-cooling deposited Ni₈₀Fe₂₀ onto spacer, implying the change of magnetic domain in transferred Ni₈₀Fe₂₀ can be quick and consistent with external magnetic field. While in the LN-cooling deposited Ni₈₀Fe₂₀, slow change with magnetic field implies the

inconsistent change of magnetic domain. Such hysteresis loops lead to the uniform or inconsistent magnetic responses in the measured magnetoresistance curves of spin valves, sharp switch between high and low resistance states can be observed in SV with transferred Ni₈₀Fe₂₀, in contrast to the slowly switched resistance states in SV with LN-cooling deposited Ni₈₀Fe₂₀. Corresponding revision has been made in the revised manuscript; please see “Such difference can also be reflected in ... devices (see Supplementary Fig. 6).” in lines 218-223, page 10.

Supplementary Fig. 5

Hysteresis loops of Al₂O₃/Ni₈₀Fe₂₀, where the Ni₈₀Fe₂₀ were fabricated by strain-restricted transfer (black) and LN-cooling evaporation (red) techniques, respectively.

Question 2-5:

4. Please introduce PC₇₁BM to the ordinary reader and reason for choosing this material.

Answer:

Thanks very much for reviewer’s kind suggestion. PC₇₁BM possesses relatively weak spin-orbit coupling due to its chemical composition of only carbon and a small number of hydrogen and oxygen, which benefits for room-temperature spin transport. Also, the excellent solubility and easily to form uniform thin film by simple spin-coating process make PC₇₁BM benefit for constructing spin valves by LN-cooling and transferred methods. Moreover, PC₇₁BM is commercially available with high purity that it can be used as we needed even in gram-scale. Accordingly, the reason for choosing PC₇₁BM as the spacer has been added and yellow-highlighted as “PC₇₁BM possesses relatively weak ... by LN-cooling and transferred methods.” in lines 234-238, page 11 of the revised manuscript.

Question 2-6:

5. Please check equation (1) to ensure its correctness. And provide the parameters (P1, P2) values.

Answer:

Thanks very much for reviewer’s comments. We have carefully checked equation (1),

and ensure it is correct referring to the previous literature. It is worth noting that, equation (1) used may be different from some literatures, since the denominator in equation (1) is R_p rather than R_{ap} (we have made this representation explicit in the revised manuscript), however, both of them are right. In fact, there are many references (Nat. Nanotechnol., 2007, 2, 216; Nat. Commun., 2019, 10, 3877; Sci. Rep., 2015, 5, 9355) use the same expression to equation (1). To make it clear, we have revised the equation (1) and the corresponding illustration, please see “Where ΔR is the difference ... magnetic alignments of FMs,” in lines 260-261, page 12 of the revised manuscript.

Moreover, the fitted P_1P_2 in LN-cooling and transfer method according to Fig. 3d are 0.03 and 0.05, respectively, which indicates a more efficient spin injection or detection in the latter devices. Accordingly, related revisions have been added and yellow-highlighted, please see “The fitted P_1P_2 in LN-cooling ... or detection in the latter devices.” in lines 267-269, page 12 in the revised manuscript.

Question 2-7:

6. As we can see from Fig.3d, why not keep the same thickness of PC71BM in LN-cooling or transfer preparation spin valves when fitting equation (1)?

Answer:

Thanks very much for reviewer’s comments. According to reviewer’s suggestions, to keep the same thickness of PC₇₁BM in LN-cooling and transfer-prepared spin valves in Fig. 3d, we have added the MR data points of transfer-prepared spin valves based on PC₇₁BM with the thicknesses of 28 nm, 38 nm and 43 nm. The revised Fig. 3d has been displayed in page 13 of the revised manuscript. Moreover, representative raw MR curves of each data point as shown in Fig. 3d are added in the Supplementary Fig. 8 (page 11 of the revised Supplementary Materials).

Fig. 3 | d, Fitting of MR data at various thicknesses of PC₇₁BM in SVs prepared via LN-cooling (blue square and line) and transfer method (red square and line).

Supplementary Fig. 8

a-c, MR curves of PC₇₁BM-based SVs prepared by LN-cooling technique, where the thicknesses of PC₇₁BM are 28 nm, 38 nm, 43 nm, respectively. **d-i**, MR curves of PC₇₁BM-based SVs prepared by transfer technique, where the thicknesses of PC₇₁BM are 28 nm, 38 nm, 43 nm, 60 nm, 78 nm, 92 nm, respectively.

Question 2-8:

7. Attenuation structure tomography in supplementary Fig.5 needs more explanation and description. Moreover, the attenuation structure tomography of the transferred Ni₈₀Fe₂₀/PC₇₁BM interface should also be provided.

Answer:

Thanks very much for reviewer's insightful suggestion. For the cases of evaporating several nanometers (such as 3 nm) Ni₈₀Fe₂₀ with LN-cooling method on PC₇₁BM layer, Ni₈₀Fe₂₀ atoms can penetrate into the PC₇₁BM layer very easily, and Ni₈₀Fe₂₀ islands is formed to present non-dense Ni₈₀Fe₂₀ film at PC₇₁BM/Ni₈₀Fe₂₀ interface. Once applying soft plasma etching, PC₇₁BM is stripped away layer by layer. As a result, a decrease in absorption is obtained with continuous tomography for the structure of PC₇₁BM/Ni₈₀Fe₂₀ prepared by LN-cooling-method (Supplementary Fig. 10a). However, the structure of PC₇₁BM/Ni₈₀Fe₂₀ (3 nm) prepared by transfer method, dense FM layer protects PC₇₁BM well, where absorption does not show attenuation (Supplementary Fig. 10b). This indicates the interface information of transferred interface is difficult to be

achieved by attenuation structure tomography technique. However, based on above analysis, it is predicted $\text{PC}_{71}\text{BM}/\text{Ni}_{80}\text{Fe}_{20}$ prepared by transfer method can effectively prevent FM metal penetration and protect PC_{71}BM layer. Otherwise, absorption should show an attenuation with continuous tomography.

According to reviewer's suggestions, more explanations and descriptions have been added and please see Supplementary Note 1 in the revised Supplementary Materials. Moreover, the attenuation structure tomography of the transferred $\text{PC}_{71}\text{BM}/\text{Ni}_{80}\text{Fe}_{20}$ interface has been provided in the Supplementary Fig. 10b of the revised version.

Supplementary Note 1: Explanation and description of attenuation structure tomography in Supplementary Fig. 10.

For the evaporation of several nanometers (such as 3 nm) $\text{Ni}_{80}\text{Fe}_{20}$ on prepared PC_{71}BM layer, the hot $\text{Ni}_{80}\text{Fe}_{20}$ atoms can penetrate into the PC_{71}BM layer very easily, where $\text{Ni}_{80}\text{Fe}_{20}$ islands are formed to present non-dense $\text{Ni}_{80}\text{Fe}_{20}$ film at $\text{PC}_{71}\text{BM}/\text{Ni}_{80}\text{Fe}_{20}$ interface. Once applying soft plasma etching, PC_{71}BM is stripped away layer by layer. As a result, a decreased absorption is obtained with continuous tomography for the structure of $\text{PC}_{71}\text{BM}/\text{Ni}_{80}\text{Fe}_{20}$ prepared by LN-cooling-method (Supplementary Fig. 10a). However, as for the structure of $\text{PC}_{71}\text{BM}/\text{Ni}_{80}\text{Fe}_{20}$ (3 nm) prepared by lamination method, absorption does not show attenuation (Supplementary Fig. 10b), meaning the lamination method can help to form high-quality $\text{Ni}_{80}\text{Fe}_{20}/\text{PC}_{71}\text{BM}$ interlayer.

Supplementary Fig. 10

a,b, Attenuation structure tomography through in-situ ultraviolet-visible tomography analysis of (a) LN-cooling prepared sample and (b) transfer prepared sample, which indicates $\text{Ni}_{80}\text{Fe}_{20}$ diffusion into the PC_{71}BM layer between 3~6 nm via LN-cooling preparation.

Question 2-9:

8. *Supplementary Fig.7 should be revised for clarity. Please add a scale bar in supplementary Fig.8.*

Answer:

Thanks very much for reviewer's comments and suggestions. The previous Supplementary Fig.7 (now it is Supplementary Fig. 13 in the revised Supplementary Materials) has been revised for clarity. The previous Supplementary Fig. 8 (now it is Supplementary Fig. 15 in the revised Supplementary Materials) is a picture, and it is not suitable to add a scale bar in this picture. We assume that the reviewer wants to let the reader know the size of the sample in the picture, as a result, we have added the sample size ($4 \times 4 \text{ cm}^2$) in the caption of Supplementary Fig. 15. Please see Supplementary Fig. 13, Fig. 15 and the caption of Supplementary Fig. 15 in the revised Supplementary Materials.

Question 2-10:

9. *Was any adhesion analysis performed on the transferred Ni₈₀Fe₂₀/PC₇₁BM? A major point that should be addressed is the generality of the polymer-assistant strain-restricted transfer technique. The readers are curious about the generality of this novel transfer technique. As for the transferred metal on various-kind channel materials, there is a lot of characterization around the magnetoresistance, but no accordingly hysteresis loops or adhesion analysis.*

Answer:

Thanks very much for reviewer's comments. We tried to peel off the transferred PS/Ni₈₀Fe₂₀ from the PC₇₁BM spacer, and the adhesion force between PS/Ni₈₀Fe₂₀ and PC₇₁BM is very large actually, which greatly exceeds the tensile strength of PS film. And in this case, we actually measured the strength of PS other than the adhesion. When we use an adhesive tape to protect and peel the PS, we found the PC₇₁BM (also happened with other spacer materials) can be peeled up from the substrate. And in this case, we actually measured the adhesion between spacer and substrate. According to the above results, the adhesion force on the transferred Ni₈₀Fe₂₀/PC₇₁BM (or Ni₈₀Fe₂₀/other spacers) is difficult to be accurately measured, but it should be larger than the PS strength ($\sim 37 \text{ MPa}$ or $7.4\text{E}4 \text{ N/m}$) or the adhesion between spacer and substrate. In fact, similar cases can also be observed in SVs based on various channel materials. Therefore, we can hardly provide the adhesion analysis of the transferred metal on various-kind channel materials.

For the generality of the polymer-assistant strain-restricted transfer technique, it is suitable for many types of spacer materials (almost all of the organic semiconductors, and most of inorganic semiconductor and some of carbon-based materials) and different surface states of spacers (smooth, rough, continuous, or discontinuous), as well as

various substrates (glass, silicon, silica and PET). Our transfer technique only requires the spontaneous electrostatic adhesion or thermal-lamination-induced adhesion between the polymer film and the spacer materials, which can be controlled and adjusted by surficial modification, changing treatment conditions and polymer materials, which possesses a wide applicability.

As for the characterizations of transferred metal on various-kind channel materials, the magnetic hysteresis loops of transferred $\text{Ni}_{80}\text{Fe}_{20}$ have been measured. Taking the PC_{71}BM -based devices for example, as shown in the Supplementary Fig. 7, the measured coercivity of transferred 50-nm-thick $\text{Ni}_{80}\text{Fe}_{20}$ onto 55-nm-thick PC_{71}BM spacer material is very small, much smaller than the switch field (~ 100 mT) of the measured MR curve in Fig. 3c of revised manuscript. Considering the high switch field of MR curve may be induced by the interface effect of $\text{PC}_{71}\text{BM}/\text{Ni}_{80}\text{Fe}_{20}$, and which is only a small part of the bulk $\text{Ni}_{80}\text{Fe}_{20}$, we transferred 5-nm-thick $\text{Ni}_{80}\text{Fe}_{20}$ electrodes onto PC_{71}BM spacer for contrast. We found the measured coercivity of 5-nm-thick $\text{Ni}_{80}\text{Fe}_{20}$ is basically consistent to the switch field of PC_{71}BM -based devices. Such influence of FM thickness on hysteresis loops is consistent with the previous published works (J. Appl. Phys., 2008, 103, 093720; Synth. Met., 2013, 173, 51-56). Moreover, 5-nm-thick $\text{Ni}_{80}\text{Fe}_{20}$ were transferred onto other spacer materials and the corresponding hysteresis loops have also been provided in Supplementary Fig. 14 of the Supplementary Materials. The corresponding contents have been added in the revised manuscript and Supplementary Materials. Please see “And the switching magnetic-field ... provided in Supplementary Fig. 7.” in lines 250-252, page 12, and “the hysteresis loops...in Supplementary Fig. 14.” in lines 355-356, page 18 of revised manuscript.

Supplementary Fig. 7

Hysteresis loops of 50-nm-thick (black line) and 5-nm-thick (red line) $\text{Ni}_{80}\text{Fe}_{20}$ onto PC_{71}BM channel materials at 300K.

Supplementary Fig. 14

a-g, Hysteresis loops of 5-nm-thick $\text{Ni}_{80}\text{Fe}_{20}$ on different channel materials, including $[\text{Au}_{25}(\text{PPh}_3)_{10}(\text{PET})_5\text{Cl}_2]^{2+}$ NCs (**a**), lithium fluoride (**b**), spin-coated TPTDPP COF (**c**), P3HT (**d**), CuPc (**e**), tips-pentacene (**f**), in-situ grown TPTDPP COF (**g**).

Response to Reviewer #3:

Question 3-1:

The submitted article deals with the formation of strain-restricted ferromagnetic electrodes, which are essential to high quality spintronic devices. A major disadvantage of the energetic metal deposition process is the possibility of the introduction of defects, disorder, etc. into non-magnetic material, resulting in a partial degradation of device performance and a lack of reproducibility. The proposed polymer-assistant strain-restricted transfer technique provides an alternative method to address the issues related to the metal deposition process, in which the electrodes are transferred onto the material of interest. The topic is rather interesting and may attract much interest from the scientists working in the field of spintronics. The experiments and results appear to substantiate the conclusions and interpretations in the submission. Prior to the acceptance of submission, some minor revisions listed below need to be considered in earnest:

Answer:

Thanks very much for reviewer's positive comments and appreciations.

Question 3-2:

1. In Figure 1a, as the stress increases from 1.5% to 1.6%, the area of the loop increases significantly. Is the material exhibiting plastic deformation, as opposed to just elastic deformation? Maybe the sudden change in the hysteresis loop indicates that the material is approaching its yield point.

Answer:

Thanks very much for reviewer's insightful comments. For the first question, in our experiments, the force on the samples have been completely released after a certain strain stretch, therefore, the residual stress may exist only when the sample is plastically deformed during stretching. And such residual stress might lead to the increased area of the hysteresis loops. In this case, we also measured the electrical characterization before and after the threshold strain (at 1.6% for Co, and at 1.8% for Ni₈₀Fe₂₀), which shows negligible changes (Supplementary Fig. 1d and 1e). So, we can confirm that the plastic deformation may happen but should be very slightly (when the stress increases from 1.5% to 1.6%). These results also imply that the magnetic property might be more sensitive to residual stress than the electrical property.

Supplementary Fig. 1d & 1e

d, Current-voltage curves of Co electrodes before (black line) and after (red line) a strain of 1.6%. **e**, Current-voltage curves of Ni₈₀Fe₂₀ electrodes before (black line) and after (red line) a strain of 1.8%.

For the second question, as the reviewer's assumption, we also think the sudden change in the hysteresis loops may indicate that the FM metal is approaching its yield point. Since the yield stage closely follows the elastic stage in the stress-strain curve of metal, when the applied strain is approaching the yield point, the sample will undergo the sudden change from elastic deformation to plastic deformation. As discussed in the previous question, plastic deformation and thus sudden change in hysteresis loops will be occurred after the threshold strain, which implies the metal is approaching its yield point. However, since the applied FM film in our paper is very thin (tens of nanometers), actually, it is difficult to verify above conjecture via the stress-strain measurements of such thin FM films.

Question 3-3:

2. Are there any wrinkles formed during the transfer process? I think heat transfer processes can cause wrinkles in the material being transferred due to the expansion and contraction of the material. Is there a stress impact associated with these wrinkles?

Answer:

Thanks very much for reviewer's comments. In fact, the peeling-off process will induce an elastic deformation in PS film, however, such temporary deformation will be recovered after peeling off process, and the PS will not wrinkle. The heat temperature to laminate the PS/FM onto the target substrate is only 106°C, at this temperature, the strain of PS film caused by thermal expansion and contraction is about 0.64%, and it is about 0.14% for Ni₈₀Fe₂₀ and 0.096% for Co. These strains are at the elastic strain zone of both PS thin film and FM electrodes, and which are lower than the threshold strain of 1.5%. Therefore, it can be concluded such strain or expansion and contraction of the materials will not cause wrinkles and thus influence the magnetic property of transferred metal.

Question 3-4:

3. *Is this transfer process repeatable? How many times can the glass-OTS/PS substrates sustain for such process including metal deposition and transfer? In the previous study (Nature volume 544, pages340–343(2017)), graphene/Ge substrate can be re-used several times for the transfer of various metal electrodes. The author should exhibit the potential of the proposed polymer-assistant strain-restricted transfer technique.*

Answer:

Thanks very much for reviewer's comments. The transfer process and the performance of the fabricated spin valve are repeatable, however, the glass-OTS/PS substrates cannot be re-used for many times (surely the glass can be re-used). In our transferred technique, the adhesion between glass and PS should be decreased to an appropriate level. So, when modifying the glass substrate with OTS, excess OTS has been used to form a weak physical interaction between glass and PS. On one hand, the OTS used for decreasing the adhesion energy between glass and PS has been exhausted after one transfer process, its effect of reducing adhesion will be obviously reduced. Therefore, the recycled glass should be re-modified by OTS when it is reused. On the other hand, for the PS film, the adhesion force between transferred PS/Ni₈₀Fe₂₀ and PC₇₁BM is so large that has already exceeded the tensile strength of PS film. Therefore, the PS film and Ni₈₀Fe₂₀ will be broken if forcibly peeling off it from the target substrate, and thus the PS film will not be of value for reuse.

However, glass-OTS/PS substrates can be obtained in low-cost way in our experiments. Firstly, the OTS modification process can be easily conducted in OTS solutions (already reported by various literatures). Secondly, PS film can be deposited onto OTS-modified glass easily by large-area blade coating. Therefore, we believe that the unable reuse of such substrates will not affect the potential of our transfer method. Corresponding illustration has been added in the Method section of the revised manuscript, please see "Moreover, the glass/OTS/PS/Ni₈₀Fe₂₀ part ... it still potential in cost saving and large-scale preparation." in lines 391-393, page 19 of the revised manuscript.

Question 3-5:

4. *Low quality images are shown in Figures 3 e, f. The author should provide more evidence to support the statement, such as high-quality cross-sectional HRTEM image for interface.*

Answer:

Thanks very much for reviewer's comments. According to the reviewer's suggestion, we have provided the high-quality cross-sectional HRTEM images of Ni₈₀Fe₂₀/PC₇₁BM interfaces prepared by LN-cooling and transfer techniques, respectively, in Supplementary Fig. 9 of the revised Supplementary Materials. It can be seen the

obvious lattice fringes in both Supplementary Fig. 9a and 9b, which can meet the requirement of reviewer's concern on high-quality TEM pictures. And the relatively rough interface of LN-cooling fabricated interface can be obviously found in contrast to the transferred fabricated interface. The corresponding contents have been added in the revised manuscript, please see "the corresponding high-resolution TEM is shown in Supplementary Fig. 9" in line 281, page 13.

Supplementary Fig. 9

a,b, Cross-sectional high-resolution transmission electron microscopy (HRTEM) images of PC₇₁BM/Ni₈₀Fe₂₀ interfaces prepared by (a) LN-cooling and (b) transfer methods. The dashed frames of the parallelogram show the lattice fringes in Ni₈₀Fe₂₀ layers.

Change List

Revised part or item	Revisions	Responding to
Main text	Lines 73, Page 4 “(about three orders of magnitudes herein)” has been added.	Reviewer 2# Question 2-3
	Lines 90-93, Page 5 “According to the inverse magnetostriction, ... (for FMs with isotropy magnetostriction,” has been added.	Reviewer 1# Question 1-3
	Line 97, Page 5 of the original manuscript “since Co with relatively high λ should be more sensitive to strain” has been deleted	Reviewer 1# Question 1-4
	Lines 102-105, Page 5 “Combined with the nearly-same conductivity ... or sample damage.” has been added.	Reviewer 1# Question 1-5
	Lines 150-151, Page 7 “(lower adhesion force between ... with the targeted substrate)” has been added.	Reviewer 2# Question 2-1-II
	Lines 157-162, Pages 7-8 “Experimentally, the adhesion differences ... is about three orders of magnitudes,” has been rewritten.	Reviewer 2# Questions 2-1-II, 2-3
	Lines 203-206, Page 10 “Note that all the AMR ... (corresponding to the left peak of curves).” has been added.	Reviewer 1# Question 1-2; Reviewer 2# Question 2-1-II
	Lines 218-223, Page 10 “Such difference can also be ... whole devices (Supplementary Fig. 6).” has	Reviewer 2# Question 2-4

	been rewritten.	
	Lines 234-238, Page 11 “PC ₇₁ BM possesses relatively ... and transferred methods.” has been added.	Reviewer 2# Question 2-5
	Lines 250-252, Page 12 “And the switching magnetic-field ... in Supplementary Fig. 7.” has been added.	Reviewer 2# Question 2-10
	Lines 260-261, Page 12 “Where ΔR is the difference ... magnetic alignments of FMs,” has been revised.	Reviewer 2# Question 2-6
	Line 265, Page 12 “, the corresponding raw MR curves are shown in Supplementary Fig. 8” has been added.	Reviewer 1# Question 1-7
	Lines 267-269, Page 12 “The fitted P_1P_2 in LN-cooling ... or detection in the latter devices.” has been added.	Reviewer 2# Question 2-6
	Line 281, Page 13 “, the corresponding high-resolution TEM is shown in Supplementary Fig. 9” has been added.	Reviewer 3# Question 3-5
	Line 283, Page 14 “Supplementary Fig. 10 and Supplementary Note 1” has been added.	Reviewer 2# Question 2-8
	Lines 287-293, Page 14 “Also, the FM atoms diffused ... seen in Supplementary Note 2.” has been added.	Reviewer 2# Question 2-1-I
	Lines 295-296, Page 14 “spin-injection and” and “ P_1P_2 and” have been added.	Reviewer 2# Question 2-1-I
	Line 322, Page 16 “20 nm” and “80 nm” have been changed into “28 nm” and “78 nm”.	Reviewer 1# Question 1-7

	Lines 323-326, Page 16 “ R_{A-Ref} of LN-cooling ... $100 \times 100 \mu\text{m}^2$ to $200 \times 500 \mu\text{m}^2$.” has been added.	Reviewer 1# Question 1-8
	Lines 355-356, Page 18 “, the hysteresis loops of transferred ... Supplementary Fig. 14.” has been added.	Reviewer 2# Question 2-10
	Lines 377-388, Pages 18-19 “The $\text{Ni}_{80}\text{Fe}_{20}$ electrodes ... can be achieved.” has been rewritten.	Reviewer 1# Questions 1-2, 1-9 Reviewer 2# Question 2-1-II
	Lines 391-393, Page 19 “Moreover, the glass/OTS/PS/ $\text{Ni}_{80}\text{Fe}_{20}$ part ... and large-scale preparation” has been added.	Reviewer 3# Question 3-4
	Lines 395-397, Page 19 “The SVs in this manuscript ... $100 \times 100 \mu\text{m}^2$ to $200 \times 500 \mu\text{m}^2$.” has been added.	Reviewer 1# Question 1-2
	Lines 423-427, Page 20 “For the measurement of strain history ... vibrating sample magnetometer (VSM).” has been added.	Reviewer 1# Questions 1-2, 1-4 Reviewer 2# Question 2-1-II
	Lines 435-436, Page 21 “The hysteresis loops of FMs ... by VSM of PPMS.” has been rewritten.	Reviewer 1# Questions 1-2, 1-4 Reviewer 2# Question 2-1-II
	Lines 441-442, Page 21 “By applying a constant bias ... can be calculated by formula (1).” has been rewritten.	Reviewer 1# Question 1-2
Formulas	Line 259, Page 12 Formula (1) has been changed into	Reviewer 2# Question 2-6

	$MR = \frac{\Delta R}{R_p} = \frac{2P_1P_2e^{-(d-d_0)/\lambda s}}{1 - P_1P_2e^{-(d-d_0)/\lambda s}}$	
	Formula (3) in Line 399 of the original manuscript has been deleted since it is same as formula (1) in Line 259 of the revised manuscript.	None
Figures	Line 195, Page 9 Fig. 2d has been revised.	Reviewer 1# Question 1-6
	Line 270, Page 13 Fig. 3d has been revised.	Reviewer 1# Question 1-7; Reviewer 2# Question 2-7
	Line 317, Page 15 Fig. 4d, 4e, 4f have been revised.	Reviewer 1# Questions 1-7, 1-8
References	Lines 527-529, Page 24 References 40 and 41 have been added.	Reviewer 2# Question 2-1-I
Acknowledgements	Line 536, Page 25 Grant No. XDB0520000 has been added.	None
Supplementary Materials	Page 2 Supplementary Note 1 has been added.	Reviewer 2# Question 2-8
	Page 3 Supplementary Note 2 has been added.	Reviewer 2# Question 2-1-I
	Page 4 Supplementary Fig. 1d, 1e and their captions have been added.	Reviewer 1# Question 1-5; Reviewer 3# Question 3-2
	Page 6 Supplementary Fig. 3 and its caption have been added.	Reviewer 2# Questions 2-1-II, 2-3
	Page 8 Supplementary Fig. 5 and its caption	Reviewer 2# Question 2-4

	have been added.	
	Page 10 Supplementary Fig. 7 and its caption have been added.	Reviewer 2# Question 2-1-III; Reviewer 2# Question 2-10
	Page 11 Supplementary Fig. 8 and its caption have been added.	Reviewer 1# Question 1-7; Reviewer 2# Question 2-7
	Page 12 Supplementary Fig. 9 and its caption have been added.	Reviewer 2# Question 2-2; Reviewer 3# Question 3-5
	Page 13 Supplementary Fig. 10b and its caption have been added.	Reviewer 2# Question 2-8
	Page 14 Supplementary Fig. 11 and its caption have been added.	Reviewer 2# Question 2-1-I
	Page 16 Supplementary Fig. 13 has been revised.	Reviewer 2# Question 2-9
	Page 17 Supplementary Fig. 14 and its caption have been added.	Reviewer 2# Questions 2-1- III, 2-10
	Page 18 The caption of Supplementary Fig. 15 has been revised.	Reviewer 1# Question 1-9; Reviewer 2# Question 2-9
	All the numbers of Supplementary Figures have been changed in turn due to the added Figures.	None
	Page 19 References 1 and 2 have been added.	Reviewer 2# Question 2-1-I

REVIEWERS' COMMENTS

Reviewer #1 (Remarks to the Author):

The authors have addressed all of my previous comments appropriately, and the manuscript is improved. This manuscript is highly evaluated also by the other reviewers. I thus recommend the publication of this manuscript in Nature Communications without further revisions.

Reviewer #2 (Remarks to the Author):

I am satisfied with the revisions made to the manuscript. The revised manuscript has been greatly improved and it can now be published.

Reviewer #3 (Remarks to the Author):

After checking the revised manuscript, I feel that all the concerns I raised have been addressed properly. Publication in Nature Communications is recommended.